# Mitigating the Noise Shift for Denoising Generative Models via Noise Awareness Guidance

**Jincheng Zhong**[1],[*] **Boyuan Jiang**[2], **Xin Tao**[2], **Pengfei Wan**[2], **Kun Gai**[2], **Mingsheng Long**[1][✉]

[1] School of Software, BNRist, Tsinghua University, China
[2] Kling Team, Kuaishou Technology, China

```
{zhongjinchengwork, jiangsutx}@gmail.com
{jiangboyuan,wanpengfei}@kuaishou.com
 mingsheng@tsinghua.edu.cn
```

### Abstract

Existing denoising generative models rely on solving discretized reverse-time SDEs or ODEs. In this paper, we identify a long-overlooked yet pervasive issue in this family of models: a misalignment between the pre-defined noise level and the actual noise level encoded in intermediate states during sampling. We refer to this misalignment as *noise shift*. Through empirical analysis, we demonstrate that noise shift is widespread in modern diffusion models and exhibits a systematic bias, leading to sub-optimal generation due to both out-of-distribution generalization and inaccurate denoising updates. To address this problem, we propose *Noise Awareness Guidance* (NAG), a simple yet effective correction method that explicitly steers sampling trajectories to remain consistent with the pre-defined noise schedule. We further introduce a classifier-free variant of NAG, which jointly trains a noise-conditional and a noise-unconditional model via noise-condition dropout, thereby eliminating the need for external classifiers. Extensive experiments, including ImageNet generation and various supervised fine-tuning tasks, show that NAG consistently mitigates noise shift and substantially improves the generation quality of mainstream diffusion models. Code is publicly available now [1].

## 1 Introduction

denoising generative models, such as diffusion models (Ho et al., 2020) and flow-based models (Lipman et al., 2023), have demonstrated remarkable scalability and achieved state-of-the-art results across a wide range of visual generation tasks, including image synthesis (Ho et al., 2020), video generation (Ho et al., 2022), and cross-modal generation (Saharia et al., 2022). The core principle of these models is to progressively recover a target sample from pure noise. At each iteration, a network processes an intermediate state, which consists of both signal and noise mixed in pre-defined proportions, and updates it to the next state according to the output and pre-defined coefficients.

During iterative sampling, the model is repeatedly applied and inevitably accumulates errors from multiple sources, including imperfect network approximation, discretization in numerical integration, and other stochastic factors. Recent studies have primarily focused on the discretization aspect, aiming to accelerate generation by reducing the number of denoising steps (Geng et al., 2025; Song et al., 2023; Lu et al., 2022), or on designing more effective diffusion architectures to increase model capacity (Peebles & Xie, 2023; Ma et al., 2024; Karras et al., 2022). Nevertheless, accumulated errors in such a complex system are unavoidable. A key manifestation of these errors is that the noise level inherently encoded in intermediate states may deviate from the pre-defined schedule. This misalignment, long overlooked by the community, is both widespread and rooted in the collective effect of diverse error sources. We refer to this phenomenon as *noise shift*, which often leads to a fundamental mismatch between training and inference in denoising networks.

In this work, we demonstrate that the noise shift manifests as a systematic drift toward larger noise levels $t'$. We conduct an empirical analysis on recent advanced diffusion models (Ma et al., 2024)

---

[*] Work done during internship at Kling Team, Kuaishou Technology
[1] Code is publicly available at: https://github.com/KlingAIResearch/noise-awareness-guidance

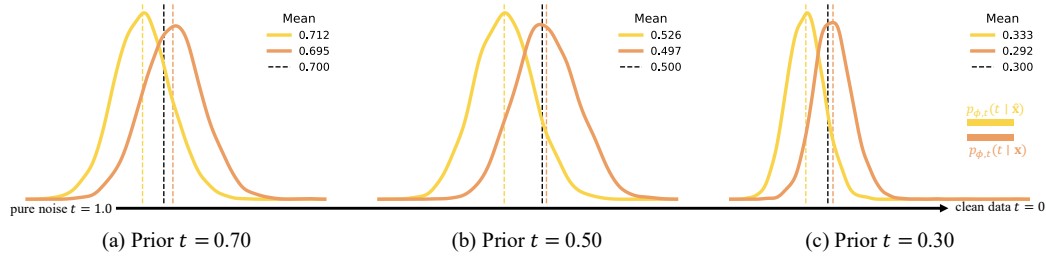

(a) Prior $t = 0.70$  (b) Prior $t = 0.50$  (c) Prior $t = 0.30$

Figure 1: **Empirical observation of *noise shift*.** Denoising generative models suffer from a training–inference misalignment, where the posterior estimation during sampling tends to lean toward larger noise levels. The yellow curves indicate the estimated probability density of the posterior $p_{\phi,t}(t \mid \hat{\mathbf{x}})$ for sampled intermediate states $\hat{\mathbf{x}}$, while the orange curves indicate the posterior $p_{\phi,t}(t \mid \mathbf{x})$ for intermediate states $\mathbf{x}$ stochastically interpolated from training data $\mathbf{x}_0 \sim p_{\text{data}}(\mathbf{x}_0)$ on ImageNet. The (a), (b), and (c) show comparisons between posterior estimates obtained at inference and training, for prior noise levels $t = 0.7$, $0.5$, and $0.3$, respectively. All density functions are estimated via kernel density estimation with 5,000 samples.

for ImageNet generation. As illustrated in Figure 1, the noise shift issue is widespread and can be directly observed using an external posterior noise-level estimator $g_{\phi}$. This observable noise shift $\delta$ indicates a clear mismatch: the actual noise encoded in intermediate states is not consistent with the pre-defined noise levels, exhibiting a systematic tendency toward larger noise levels $t' = t + \delta$. To quantify noise shift, we compare the posterior estimation $g_{\phi}(t \mid \hat{\mathbf{x}})$ of intermediate states during sampling with the posterior estimation $g_{\phi}(t \mid \mathbf{x}_t)$ of intermediate states from the forward process in training, along with the reference of the corresponding pre-defined prior $t$.

This misalignment can lead to sub-optimal results in two ways: 1) noise shift introduces out-of-distribution generalization issues, since the trained model is applied to a shifted intermediate state $\mathbf{s}_{\theta}(\mathbf{x}_{t+\delta}, t)$ rather than the intended $\mathbf{s}_{\theta}(\mathbf{x}_t, t)$. 2) noise shift causes sub-optimal denoising operations, as the next state is computed using inaccurate pre-defined coefficients.

To address this issue, we propose *Noise Awareness Guidance* (NAG), a novel guidance correction approach designed to mitigate the noise shift phenomenon. The key idea of NAG is to enable denoising models to recognize the inherent noise level of a given intermediate state during sampling and to generate a guidance signal that steers shifted samples back toward the accurate pre-defined noise level. However, as discussed in prior works (Ho & Salimans, 2021; Dhariwal & Nichol, 2021), gradient-based guidance signals that rely on external classifiers suffer from several drawbacks, including vulnerability to adversarial-like gradient manipulation, complex training pipelines, and the need for additional costly training on noisy inputs. Inspired by the success of classifier-free guidance (CFG) (Ho & Salimans, 2021), we further propose a classifier-free variant of NAG. Instead of relying on the gradient of a separately trained noise estimator, classifier-free NAG combines the score estimates of a noise-conditional diffusion model with those of a jointly trained noise-unconditional model. This approach removes the dependency on external classifiers by applying noise-condition dropout during training.

Empirically, we show that NAG substantially alleviates the noise shift issue, consistently leading to significant improvements in the generation quality of mainstream denoising generative models. Our comprehensive evaluations are conducted across two widely used base models: DiT (Peebles & Xie, 2023) for diffusion models and SiT (Ma et al., 2024) for flow-based models. To demonstrate both the effectiveness and generality of NAG, our evaluations cover two mainstream use cases of modern denoising generative models: 1) We show that NAG can be directly incorporated into DiT and SiT to improve ImageNet conditional generation, highlighting that foundation model development can benefit from our approach. 2) We conduct supervised fine-tuning experiments on small downstream datasets, verifying the effectiveness of NAG in supervised fine-tuning scenarios.

Overall, our contributions can be summarized as follows:

- We identify the noise shift issue, which is widespread in existing denoising generative models but has long been overlooked. Through empirical analysis with an external noise estimator on ImageNet generation tasks, we reveal the severity of this issue.

- We propose a novel and concise approach, *Noise Awareness Guidance* (NAG), to mitigate the noise shift issue. We further introduce its classifier-free variant, which can be more easily incorporated into mainstream denoising generative models.

- We conduct comprehensive experiments validating the effectiveness and generality of NAG, providing strong evidence that it mitigates the noise shift issue and leads to significant improvements in both ImageNet generation and supervised fine-tuning tasks.

## 2 PRELIMIARY

We begin by reviewing denoising generative models under the unified framework of *stochastic interpolants* (Albergo et al., 2023). Throughout this section, we adopt the notation of Ma et al. (2024). Both diffusion and flow-based models can be understood as stochastic processes that gradually transform a noise sample from simple prior distributions, typically a standard Gaussian $\epsilon \sim \mathcal{N}(\mathbf{0}, \mathbf{I})$, into a data sample from the complex target distribution $\mathbf{x}_0 \sim p_{\text{data}}(\mathbf{x}_0)$.

**Forward process.** Let $\mathbf{x}_0 \sim p_{\text{data}}(\mathbf{x}_0)$ be a sample from the data distribution. We define a continuous-time stochastic interpolant over $t \in [0, T]$:

$$\mathbf{x}_t = \alpha_t \mathbf{x}_0 + \sigma_t \epsilon, \quad \alpha_0 = \sigma_T = 1, \quad \alpha_T = \sigma_0 = 0, \tag{1}$$

where $\alpha_t$ is monotonically decreasing and $\sigma_t$ is monotonically increasing (Lipman et al., 2023; Ma et al., 2024). This formulation interpolates smoothly between the clean data ($t = 0$) and pure noise ($t = T$).

**Probability flow ODE.** Given the forward process, the dynamics of $\mathbf{x}_t$ can be equivalently described by a *probability flow ordinary differential equation* (PF ODE):

$$\dot{\mathbf{x}}_t = \mathbf{v}(\mathbf{x}_t, t), \tag{2}$$

where the velocity field is given by

$$\mathbf{v}(\mathbf{x}, t) = \mathbb{E}[\dot{\mathbf{x}}_t \mid \mathbf{x}_t = \mathbf{x}] = \dot{\alpha}_t \, \mathbb{E}[\mathbf{x}_0 \mid \mathbf{x}_t = \mathbf{x}] + \dot{\sigma}_t \, \mathbb{E}[\epsilon \mid \mathbf{x}_t = \mathbf{x}]. \tag{3}$$

In practice, the velocity is parameterized by a neural network $\mathbf{v}_\theta(\mathbf{x}, t)$, trained with the objective

$$\mathcal{L}_{\mathbf{v}}(\theta) := \mathbb{E}_{\mathbf{x}_0, \epsilon, t} \left[ \left\| \mathbf{v}_\theta(\mathbf{x}_t, t) - \dot{\alpha}_t \mathbf{x}_0 - \dot{\sigma}_t \epsilon \right\|^2 \right]. \tag{4}$$

Since the ODE solution at time $t$ matches the marginal distribution $p_t(\mathbf{x})$ of $\mathbf{x}_t$, samples can be generated by integrating Equation 2 backward from $\mathbf{x}_T = \epsilon \sim \mathcal{N}(\mathbf{0}, \mathbf{I})$ using standard ODE solvers.

**Reverse-time SDE.** Equivalently, the marginals $p_t(\mathbf{x})$ are consistent with the reverse-time *stochastic differential equation* (SDE):

$$d\mathbf{x}_t = \mathbf{v}(\mathbf{x}_t, t) \, dt - \tfrac{1}{2} w_t \mathbf{s}(\mathbf{x}_t, t) \, dt + \sqrt{w_t} \, d\hat{\mathbf{w}}_t, \tag{5}$$

where $\hat{\mathbf{w}}_t$ is a reverse-time Wiener process, $w_t > 0$ is a diffusion coefficient, and $\mathbf{s}(\mathbf{x}, t) = \nabla_{\mathbf{x}} \log p_t(\mathbf{x})$ is the score function. The score can be expressed either as a conditional expectation or equivalently in terms of the velocity field:

$$\mathbf{s}(\mathbf{x}, t) = -\sigma_t^{-1} \, \mathbb{E}[\epsilon \mid \mathbf{x}_t = \mathbf{x}] = -\sigma_t^{-1} \frac{\alpha_t \mathbf{v}(\mathbf{x}, t) - \dot{\alpha}_t \mathbf{x}}{\dot{\alpha}_t \sigma_t - \alpha_t \dot{\sigma}_t}. \tag{6}$$

Thus, data can also be generated by solving Equation 5 with the same velocity model $\mathbf{v}_\theta(\mathbf{x}, t)$.

**Conditional generation** Let $p_t(\mathbf{x} \mid \mathbf{y})$ is the density that $\mathbf{x}_t$ is condtioned on some variable $\mathbf{y}$. If $p_t(\mathbf{y} \mid \mathbf{x})$ is known, we can sample from $p_t(\mathbf{x} \mid \mathbf{y})$ by solving a conditional reverse-time SDE where the conditional score defined as:

$$\mathbf{s}(\mathbf{x}, t \mid \mathbf{y}) = \nabla_{\mathbf{x}} \log p_t(\mathbf{x} \mid \mathbf{y}) = \nabla_{\mathbf{x}} \log p_t(\mathbf{x}) + \nabla_{\mathbf{x}} \log p_t(\mathbf{y} \mid \mathbf{x}). \tag{7}$$

In practice, we can build a seperate neural network to model $p_t(\mathbf{y} \mid \mathbf{x})$ on noisy data, following classifier guidance (Dhariwal & Nichol, 2021; Song et al., 2020). Note that $p_t(\mathbf{y} \mid \mathbf{x}) \propto p_t(\mathbf{x} \mid$

$\mathbf{y})p_t^{-1}(\mathbf{x})$, we can derive the classifier-free guidance sampling (Ho & Salimans, 2021). Empirically, classifier-free guidance achieves significant performance.

For simplicity, we primarily consider the linear interpolant with $T = 1$, $\alpha_t = 1 - t$, and $\sigma_t = t$, following Ma et al. (2024). Nevertheless, our analysis extends naturally to other formulations such as DDPM (Ho et al., 2020), which employ discretized dynamics, alternative schedules $(\alpha_t, \sigma_t)$, or different model parameterizations.

## 3   NOISE SHIFT ISSUE IN THE DENOISING PROCESS

We identify a misalignment between the training distribution $p_t(\mathbf{x})$, obtained from clean data samples $\mathbf{x}_0 \sim p_{\text{data}}(\mathbf{x}_0)$, and the intermediate distribution $p_t(\hat{\mathbf{x}})$ encountered during the numerical solution of the SDE or ODE. Conceptually, this misalignment can be diagnosed by comparing the posterior $p_t(t \mid \mathbf{x})$ inferred from perturbed states with the pre-defined prior $p(t)$.

In practice, accumulated errors $\mathbf{e}$ from multiple sources—such as imperfect network approximation, discretization error, and other modeling inaccuracies—can be viewed as an additional Gaussian perturbation applied to $\mathbf{x}_t$, where $\hat{\mathbf{x}}_t = \mathbf{x}_t + \mathbf{e}$, where $\mathbf{e} \sim \mathcal{N}(\mathbf{0}, \sigma_e^2\mathbf{I})$. This perturbation increases the effective variance from $\sigma_t^2$ to $\sigma_t^2 + \sigma_e^2$, making the perturbed state behave as if it were sampled at a shifted noise level $t' = t + \delta$, where

$$\sigma_{t+\delta}^2 = \sigma_t^2 + \sigma_e^2. \tag{8}$$

We refer to the discrepancy $\delta = t' - t$ as the *noise shift*.

**Statement 1 (Relation between noise shift and additive error).** *Given the forward process defined in Equation 1, consider an additive error $\mathbf{e} \sim \mathcal{N}(\mathbf{0}, \sigma_e^2\mathbf{I})$. When the error variance $\sigma_e^2$ is small, the shift $\delta$ admits a first-order approximation:*

$$\delta \approx \frac{\sqrt{\sigma_t^2 + \sigma_e^2} - \sigma_t}{\dot{\sigma}_t}, \tag{9}$$

*where $\dot{\sigma}_t = d\sigma_t/dt$. (See Appendix  A for full derivations.)*

Intuitively, Statement  1 shows that accumulated errors push the effective variance in $\hat{\mathbf{x}}_t$ toward a later noise level $t' = t + \delta$, where $\delta > 0$, causing a systematic bias. For example, in the linear interpolation case $\sigma_t = t$, the shift reduces to $\delta = \sqrt{\sigma_t^2 + \sigma_e^2} - \sigma_t$, illustrating that perturbed states tend to be interpreted as noisier than intended. Although based on simplified assumptions, this analysis qualitatively captures the nature of noise shift in practical denoising processes.

**Empirical analysis.**   To better illustrate the noise shift issue, we conduct empirical simulations on ImageNet at $256 \times 256$ resolution using the pre-trained SiT-XL/2 model, which was trained for 1,400 epochs Previous studies (Sun et al., 2025; Stahl et al., 2000) suggest that for high-dimensional data such as images, the posterior $p_t(t \mid \mathbf{x})$ concentrates sharply (similar to a Dirac delta), making the noise level $t$ encoded in $\mathbf{x}$ reliably estimable. Motivated by this, we train a noise estimator $g_\phi(t \mid \mathbf{x})$ on the ImageNet $256 \times 256$ dataset[2].

Empirical comparisons between the estimated posterior distributions $p_{\phi,t}(t \mid \hat{\mathbf{x}})$ are shown in Figure  1. Consistent with Statement  1, we observe that the estimated posterior distribution $p_{\phi,t}(t \mid \hat{\mathbf{x}})$ (yellow curve) shifts toward larger values of the pre-defined prior $t$, demonstrating that the noise shift phenomenon is widespread in the denoising stage. Additionally, the orange curve shows the posterior estimation on samples generated from ImageNet through the forward process in Equation 1, serving as evidence of the accuracy of $g_\phi$ on ground-truth intermediate states.

In particular, intermediate states with mid-level noise exhibit substantial systematic overestimation by $g_\phi$, highlighting a clear misalignment between the training and inference distributions. Further results at more noise levels $t$ can be found in the Appendix D.

**The effect of noise shift $\delta$.**   While our empirical analysis is constrained by the accuracy of the noise estimator, the observed noise shift $\delta$ can still be regarded as a sufficient but not necessary condition for indicating sub-optimal behavior in the denoising stage.

---

[2]Implementation details of the noise estimator are provided in the Appendix B.2

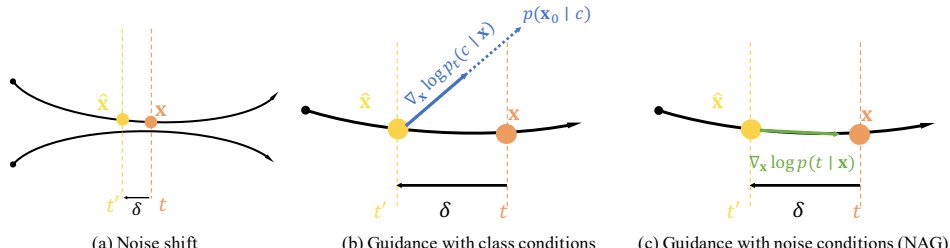

(a) Noise shift · (b) Guidance with class conditions · (c) Guidance with noise conditions (NAG)

Figure 2: **Conceptual comparison of guidance behaviors based on class information and noise awareness.** (a) A conceptual example of noise shift, where $\hat{\mathbf{x}}_t$ is drifted to a larger noise level by $\delta$. (b) Class-conditional guidance pushes the trajectory toward regions aligned with the class condition $c$. (c) Noise-aware guidance instead pushes $\hat{\mathbf{x}}_t$ toward the position better aligned with the intended noise level $t$ from the pre-defined prior. NAG explicitly targets the noise shift issue.

This pervasive noise shift affects the entire sampling trajectory in two primary ways: 1) The learned velocity field $\mathbf{v}_\theta(\mathbf{x}, t)$ suffers from out-of-distribution errors, since the model operates on perturbed intermediate states with shifted noise levels $\delta$. If the noise-conditioned network $\mathbf{v}_\theta(\mathbf{x}, t)$ satisfies a Lipschitz condition in $\mathbf{x}$, the resulting model error can be bounded by $L_\mathbf{x}\|\mathbf{e}\|$, where $L_\mathbf{x}$ is the Lipschitz constant. 2) The misalignment in $t$ introduces errors in the SDE coefficients $\alpha_t$ and $\sigma_t$ during reverse-time integration. Consequently, the denoising process becomes sub-optimal under the influence of noise shift.

As discussed above, $\delta$ can be interpreted as a collection of errors originating from various sources, making it unrealistic to eliminate completely. Notably, reducing $\delta$ to zero is not a sufficient condition for generating better images. For instance, if an intermediate sample corresponds to an image that is entirely out of distribution, generation will still fail due to the limited capability of the model. Nevertheless, since the existence of noise shift always induces some degree of misalignment, our qualitative findings provide valuable insights into the design of corrective methods.

## 4 Noise Awareness Guidance

In this section, we introduce the core concept of *Noise Awareness Guidance* (NAG), which directly addresses the noise shift issue. We interpret the shift $\delta$ as the misalignment between the sampled state $\hat{\mathbf{x}}_t$ and its intended noise condition $t$. Inspired by conditional guidance methods (Dhariwal & Nichol, 2021; Song et al., 2020), we propose a mechanism that explicitly steers the sampling trajectory to remain consistent with the pre-defined noise schedule. Our key insight is that by reinforcing the conditioning on $t$, the posterior $p_t(t \mid \hat{\mathbf{x}})$ along the reverse-time SDE (or ODE) trajectory remains closer to the pre-defined $t$, thereby mitigating the noise shift $\delta$.

**Noise awareness guidance.** The noise-conditional score can be written as

$$\mathbf{s}(\mathbf{x} \mid t) = \nabla_\mathbf{x} \log p_t(\mathbf{x} \mid t) = \nabla_\mathbf{x} \log p_t(\mathbf{x}) + \nabla_\mathbf{x} \log p_t(t \mid \mathbf{x}). \tag{10}$$

Analogous to Equation 7, if $p_t(t \mid \mathbf{x})$ were available, we could sample from $p_t(\mathbf{x} \mid t)$ by solving the conditional reverse-time SDE in Equation 10. As discussed in Section 3, the posterior $p_t(t \mid \mathbf{x})$ can be reliably estimated from a noisy data point $\mathbf{x}_t$. Intuitively, we can guide the sampling trajectory with $\nabla \log g_\phi(t \mid \mathbf{x})$ as the guidance signal, where $g_\phi$ is the posterior estimator model in Section 3. Since it relies on being aware of the accurate noise level encoded in an intermediate state. We refer to this approach as *Noise Awareness Guidance* (NAG). As the gradient $\nabla \log g_\phi(t \mid \mathbf{x})$ is provided by an external posterior estimator $g_\phi$, we call this formulation *classifier-based* NAG.

**Classifier-free noise awareness guidance.** Despite its effectiveness, classifier-based NAG inherits the drawbacks of classifier guidance (Dhariwal & Nichol, 2021; Song et al., 2020), including the high computational cost of training an external posterior estimator for $t$, increased pipeline complexity, and the risk of adversarial-like behavior in explicit classifiers. To address these issues, we extend the idea of classifier-free guidance (CFG) (Ho & Salimans, 2021) to NAG.

Noting that $p_t(t \mid \mathbf{x}) \propto p_t(\mathbf{x} \mid t)/p_t(\mathbf{x})$, we can utilize a score mixture to approximate the gradient of an implicit noise predictor as

$$\mathbf{s}^{w_{\text{nag}}}(\mathbf{x} \mid t) = (w_{\text{nag}} + 1)\,\mathbf{s}(\mathbf{x} \mid t) - w_{\text{nag}}\,\mathbf{s}(\mathbf{x}), \tag{11}$$

where $w_{\text{nag}}$ is the guidance parameter for NAG. Importantly, modern denoising models already accept the noise level $t$ along with the intermediate state $\mathbf{x}$, inherently defining the conditional score $\mathbf{s}(\mathbf{x} \mid t)$. Thus, we only need access to the unconditional score $\mathbf{s}(\mathbf{x})$, without explicitly training a separate noise-level predictor. To implement NAG, we follow the training strategy of CFG: during training, the noise condition $t$ is randomly dropped with a fixed probability, allowing the model to share weights between conditional and unconditional objectives.

**Discussion and relation to CFG.**    The mechanism of NAG can be intuitively understood in analogy to CFG. From the perspective of conditional generation, sampling without NAG corresponds to relying solely on the conditional score model. By strengthening the conditioning on $t$, NAG guides the trajectory toward lower-temperature regions where the model produces higher-confidence samples, ensuring that each intermediate state remains aligned with its intended noise level.

As illustrated in Figure 2, the noise-level conditioning axis introduced by NAG is orthogonal to the conditional axis of CFG, providing complementary control over the sampling process. It is worth noting that because the noise shift $\delta$ arises from various sources, CFG empirically mitigates it to some extent, as it biases sampling toward lower-temperature regions where models are more confident. However, compared to this indirect effect of CFG, NAG directly targets the reduction of $\delta$ and thereby constructs improved sampling trajectories. Figure 4 visualizes the mitigating effect on noise shift by different methods.

## 5 EXPERIMENTS

In this section, we present a comprehensive empirical analysis to demonstrate the effectiveness and generality of NAG. Our study considers two settings: (1) standard ImageNet generation benchmarks (Section 5.1) and (2) supervised fine-tuning off-the-shelf models on small, fine-grained datasets (Section 5.2). These experiments provide evidence of NAG's compatibility with two widely used scenarios: large-scale foundation model training and supervised fine-tuning. Section 5.3 presents more discussion on empirical analysis of noise shift $\delta$.

### 5.1 NAG FOR IMAGENET GENERATION

**Implementation details.**    Our experiments are conducted on two representative variants of denoising generative models: DiTs (Peebles & Xie, 2023) for diffusion-based models and SiTs (Ma et al., 2024) for flow-based models. We faithfully follow the experimental setups described in the DiT and SiT papers, unless otherwise specified. All experiments are performed at a resolution of $256 \times 256$ (denoted as ImageNet $256 \times 256$), where $32 \times 32 \times 4$ latent vectors are obtained using the pre-trained Stable Diffusion VAE tokenizer (Rombach et al., 2022). For model configurations, we adopt the S/2, B/2, L/2, and XL/2 variants introduced in the DiT and SiT papers (Peebles & Xie, 2023; Ma et al., 2024), all of which process inputs with a patch size of 2. For experiments trained from random initialization, we train for 80 epochs and apply a 10% dropout probability on the noise conditions. Due to computational limitations, evaluations on fully converged XL/2 models are instead conducted by fine-tuning for an additional 10 epochs on off-the-shelf checkpoints pre-trained for 1,400 epochs with 20% noise dropout. Additional experimental details are provided in Appendix B.

**Evaluation.**    For experiments with DiT, we follow the default setup using 250 DDPM sampling steps (Peebles & Xie, 2023). For SiT, consistent with its original setup, we always adopt the SDE–Euler–Maruyama sampler with 250 sampling steps (Ma et al., 2024). For experiments across different architectures of DiTs and SiTs, we report the Fréchet Inception Distance (FID) (Heusel et al., 2017) computed with 10,000 samples. For converged results, to enable direct comparison with the original papers, we report FID, precision (Prec.), and recall (Rec.) (Kynkäänniemi et al., 2019) computed with 50,000 samples by default.

**Comparison.**    Figure 3 presents the results of training DiTs and SiTs from scratch across various architectures. The results show that NAG consistently brings substantial improvements over the

Table 1: **Converged comparsions on ImageNet 256 × 256 with DiT-XL/2 and SiT-XL/2.** We fine-tune off-the-shelf DiT-XL/2 and SiT-XL/2 checkpoints for an additional 10 epochs to support NAG sampling, with and without classifier-free guidance (CFG), following the setup in the original papers (Peebles & Xie, 2023; Ma et al., 2024). All metrics are reported on 50k generated images.

| Model | Training Epochs | Generation w/o CFG | | | Generation w/ CFG | | |
|---|---|---|---|---|---|---|---|
| | | FID | Prec. | Rec. | FID | Prec. | Rec. |
| DiT-XL/2 (Peebles & Xie, 2023) | 1400 | 9.62 | 0.67 | **0.67** | 2.27 | **0.83** | 0.57 |
| +NAG (ours) | 10+(1400*) | **2.59** | **0.79** | 0.60 | **2.14** | 0.80 | **0.61** |
| SiT-XL/2 (Ma et al., 2024) | 1400 | 8.61 | 0.68 | **0.67** | 2.06 | **0.82** | 0.59 |
| +NAG (ours) | 10+(1400*) | **2.26** | **0.75** | 0.66 | **1.72** | 0.77 | **0.66** |

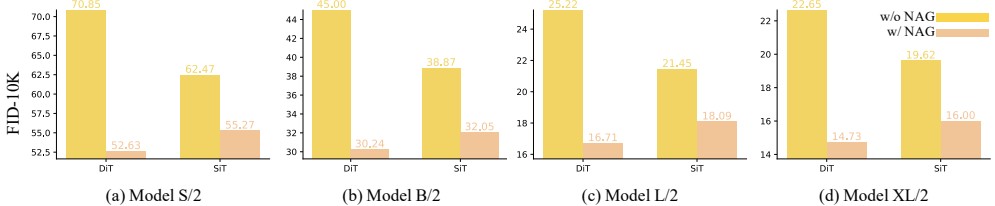

(a) Model S/2          (b) Model B/2          (c) Model L/2          (d) Model XL/2

Figure 3: **FID comparison of vanilla DiTs and SiTs** on ImageNet 256 × 256 after 80 epochs of training. Classifier-free guidance (CFG) is not used. All metrics are computed with 10K samples.

baselines. An interesting observation is that DiTs benefit more from NAG than SiTs when trained for 80 epochs. This may arise from the different training schedules: the DDPM-style setup used in DiTs could lead to better training of the noise-unconditional branch, thereby providing a more accurate guidance direction for NAG. Notably, for extensively pre-trained models, it is sufficient to fine-tune only the noise-unconditional branch at a small fraction of the original cost (e.g., 10% additional epochs, approximately 0.7% of the full 1,400-epoch pre-training cost) to enable the model to apply NAG. Remarkably, using NAG alone allows the model to achieve generation quality close to that of a CFG-guided model. Moreover, when combined with CFG, NAG continues to provide additional improvements, demonstrating that its mechanism is complementary and orthogonal to CFG.

## 5.2 NAG FOR SUPERVISED FINE-TUNING

**Implementation Details.** Supervised fine-tuning of an off-the-shelf pre-trained checkpoint to a new domain is a fundamental task in generative modeling. To further demonstrate the general effectiveness of NAG, we conduct supervised fine-tuning evaluations following the setups in Zhong et al. (2025; 2024). Specifically, we evaluate NAG on fine-tuning DiT-XL/2[3] across seven well-established fine-grained downstream datasets: Food101 (Bossard et al., 2014), SUN397 (Xiao et al., 2010), DF20-Mini (Picek et al., 2022), Caltech101 (Griffin et al., 2007), CUB-200-2011 (Wah et al., 2011), ArtBench-10 (Liao et al., 2022), and Stanford Cars (Krause et al., 2013). We fine-tune for 24,000 steps with a batch size of 32 at 256 × 256 resolution for each task. The compared baselines include vanilla generation, generation with classifier-free guidance (CFG), and Domain Guidance (DoG) (Zhong et al., 2025). Notably, DoG is a guidance method specifically designed for fine-tuning scenarios. To demonstrate both the fundamental effect and generality of NAG, we directly apply it on top of these baselines without any modifications, except for introducing noise-dropout training to support the noise-unconditional branch. Detailed implementation information is provided in Appendix B.

**Evaluations.** Following the setup in (Zhong et al., 2025), all results are generated with 50 DDPM sampling steps. and the FIDs are computed with 10,000 samples.

**Results.** The FID comparisons across various fine-tuning tasks are summarized in Table 2. The results indicate that NAG is highly general and exhibits strong compatibility across different baselines,

---

[3]https://dl.fbaipublicfiles.com/DiT/models/DiT-XL-2-256x256.pt

Table 2: **FID Comparisons on fine-tuning tasks** with pre-trained DiT-XL-2-256x256.

| Dataset / Method | Food | SUN | Caltech | CUB Bird | Stanford Car | DF-20M | ArtBench | Average FID |
|---|---|---|---|---|---|---|---|---|
| Fine-tuning (w/o CFG) | 16.04 | 21.41 | 31.34 | 9.81 | 11.29 | 17.92 | 22.76 | 18.65 |
| + NAG (ours) | 11.18 | 14.95 | 24.32 | 5.68 | 5.92 | 14.79 | 19.22 | 13.72 |
| Fine-tuning (with CFG) | 10.93 | 14.13 | 23.84 | 5.37 | 6.32 | 15.29 | 19.94 | 13.69 |
| + NAG (ours) | **5.78** | 8.81 | **21.87** | 3.52 | **3.91** | 12.55 | 15.69 | 10.31 |
| Fine-tuning (with DoG) | 9.25 | 11.69 | 23.05 | 3.52 | 4.38 | 12.22 | 16.76 | 11.55 |
| + NAG (ours) | 6.45 | **8.24** | 21.88 | **3.41** | 4.21 | **11.38** | **14.80** | **10.05** |

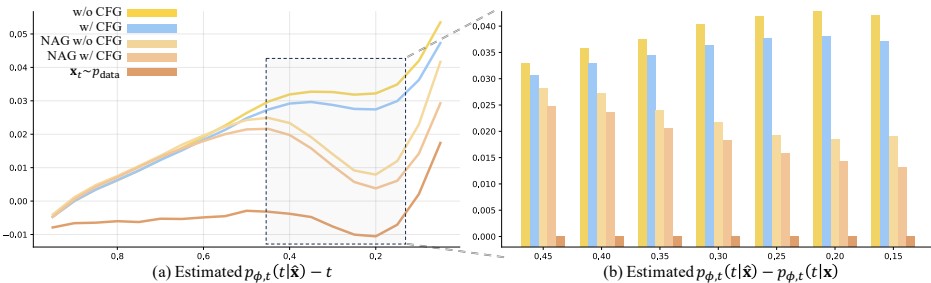

Figure 4: **Comparisons of the estimated posterior** $p_\phi(t \mid \mathbf{x})$ on ImageNet $256 \times 256$ with a converged SiT/XL-2 model. (a) Noise shift across the entire sampling process, computed as the difference between the estimated posterior $p_\phi(t \mid \hat{\mathbf{x}})$ and the pre-defined prior $t$. The visualization shows that noise shift $\delta$ becomes increasingly severe as sampling progresses. (b) Noise shift measured between the estimated $p_\phi(t \mid \hat{\mathbf{x}})$ and $p_\phi(t \mid \mathbf{x})$, where $\mathbf{x}$ is generated from real data. This comparison reflects the training–inference misalignment while accounting for the inherent inaccuracy of $g_\phi$.

benchmarks, and guidance approaches. Consistent with the ImageNet results, NAG alone achieves performance comparable to sampling with CFG. Furthermore, Table 2 shows that both CFG-guided sampling and DoG-guided sampling can be substantially improved by NAG. This broad compatibility highlights that the noise shift issue is indeed widespread in denoising-based generation, and that NAG, by directly addressing this issue, can consistently improve generation quality across various baselines. Notably, Domain Guidance (DoG) (Zhong et al., 2025), a CFG variant specifically designed for supervised fine-tuning, also benefits from NAG, with significant improvements observed in Table 2.

### 5.3 Empirical Observations of Noise Shift with NAG

We present a detailed empirical analysis based on the estimator $g_\phi$, as an expansion beyond Section 3.

As the sampling process progresses, the noise shift can be divided into two stages. In the first stage, the shift increases steadily until it reaches a threshold (e.g., when the signal-to-noise ratio is around 1). In the second stage, the shift plateaus, remaining relatively stable as the actual noise level decreases from 0.5 to 0. When intermediate states approach the data distribution at very low noise levels, the estimated noise shift relative to the pre-defined prior $t$ tends to be overestimated. This occurs because $g_\phi$ applied to intermediate states $\mathbf{x}$ generated from real data suffers from larger estimation errors due to its limited capability in this regime. This overestimate can be viewed in Figure 4(a) and released by mean normalization in Figure 4(b).

As shown in Figure 4, NAG primarily influences the sampling process when the signal-to-noise ratio is larger than 1 (roughly $t \approx 0.5$), effectively reducing the noise shift in this range. In contrast, its effect is less pronounced in the early denoising stage, where the signal-to-noise ratio is low. Figure 5 further illustrates that NAG shifts the density of intermediate states toward the posterior $p_{\phi,t}(t \mid \mathbf{x})$ estimated from real data, and hence closer to the pre-defined prior $t$.

Classifier-free guidance (CFG) (Ho & Salimans, 2021) is known to steer the sampling trajectory toward low-temperature regions associated with the target class, thereby producing higher-quality samples within high-confidence regions. This can be interpreted as a reduction of model fitting errors. Since noise shift $\delta$ reflects the accumulation of errors from multiple sources, CFG also reduces noise shift to some extent (as observed in Figure 1(a–b)). However, its effect remains indirect and limited,

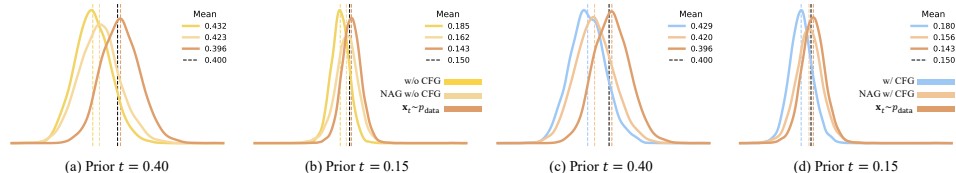

(a) Prior $t = 0.40$ (b) Prior $t = 0.15$ (c) Prior $t = 0.40$ (d) Prior $t = 0.15$

Figure 5: **Empirical observations of NAG mitigating the noise shift $\delta$.** (a–b) Effects of NAG without interference from CFG. (c–d) Compatibility of NAG under CFG, showing that NAG addresses the noise shift directly, rather than relying on the indirect effects of CFG.

as CFG primarily mitigates errors along the class-conditional dimension. In contrast, Figures 4 and 5(c–d) demonstrate that NAG can be directly applied on top of CFG-guided models, substantially reducing the remaining noise shift.

It is important to clarify that eliminating the estimated noise shift $\delta$ is not a sufficient condition for achieving optimal generation, since potential pitfalls may lie in the imperfect accuracy of the noise estimator or in other complex factors. Nevertheless, the presence of a distinguishable noise shift during sampling is a sufficient condition for sub-optimal generation. This observation motivates us to address the noise shift issue directly.

## 5.4 ABLATION ANALYSIS

The main ablation analysis results are shown in Figure 6.

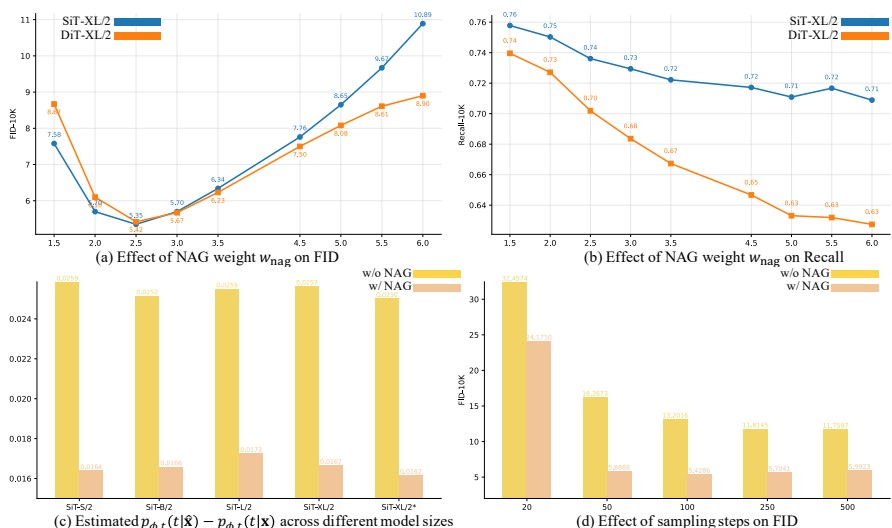

Figure 6: **Sensitivity analysis of Noise Awareness Guidance (NAG).**

**NAG weight $w_{nag}$.** We conduct analysis on how $w_{nag}$ influences the sampling results with NAG, as shown in Figure 6(a–b). As a guidance-based technique, NAG shows a similar effect to CFG; namely, with increasing $w_{nag}$, the diversity will be sacrificed, which can be seen as a form of temperature controlling in denoising generation, even though NAG does not incorporate any class information.

**Can better fitness of the denoising model reduce noise shift?** An intuitive question is that, since model prediction errors contribute to part of the noise shift, can we mitigate the noise shift by building stronger networks, such as increasing model size or training for many more iterations? Unfortunately, Figure 6(c) shows that the empirical noise shift always falls within a relatively stable range and cannot be significantly reduced by increasing the fitness of the model. Note that SiT-XL/2* is widely believed to be a convergent model with 1400 epochs of training on ImageNet256, yet it still suffers from the noise shift issue.

**Sampling steps.** Although NAG does not directly optimize the sampling steps, Figure 6(d) still shows that we can effectively save sampling steps after applying NAG. Notably, although NAG introduces a double forward pass to compute the guidance signal, we can use only around one-fifth of the sampling steps (50 steps compared to 250 steps) to achieve comparable sampling results.

## 6 RELATED WORK

**Denoising generative models.** Denoising generative models, including diffusion models and flow-based models (Ho et al., 2020; Song & Ermon, 2019; Song et al., 2020; Lipman et al., 2023), generate high quality samples from pure noise through an iterative denoising process. Recent progress in this field has primarily focused on noise schedules (Nichol & Dhariwal, 2021; Karras et al., 2022), training objectives (Salimans & Ho, 2021), and model architectures (Peebles & Xie, 2023; Ma et al., 2024), which aim to reduce approximation errors caused by limited model capacity. Another important direction is the development of faster denoising methods with fewer iterative steps, such as high order solvers (Bao et al., 2022; Lu et al., 2022) and improved interval modeling (Frans et al., 2025; Geng et al., 2025; Song et al., 2023). These works primarily address numerical errors introduced by discretized integration. In contrast, most prior studies have focused on eliminating specific sources of error. In this paper, we instead highlight a pervasive issue, namely noise shift, and demonstrate how addressing it alleviates the persistent sub optimality in the generation process.

**Training–inference misalignment.** Training–inference misalignment is a fundamental challenge that has accompanied the development of generative modeling. Modern generative models suffer from this issue severely, particularly due to their multistep sampling nature. In autoregressive models, each token is generated conditioned on previous model predictions, and accumulated errors propagate throughout the sampling chain, a well-known problem referred to as *exposure bias* (Bengio et al., 2015; Ranzato et al., 2015; Schmidt, 2019; Zhang et al., 2025). Recent works have investigated analogous misalignment phenomena in diffusion models (Ning et al., 2023a;b). Specifically, Ning et al. (2023b) aims to reduce generalization error under misalignment by improving the model's Lipschitz continuity, while Li et al. (2023) manipulates the sampling schedule to mitigate the mismatch. Ning et al. (2023a) further proposes a training-free epsilon scaling method that rescales the removed noise at intermediate states. Compared to these works, our formulation of *noise shift* provides an empirical and objective perspective for quantifying the misalignment, and it naturally motivates the design of NAG. Abuduweili et al. (2025) also adopts a noise-level perspective to identify misalignment; however, their analysis remains primarily theoretical, lacks clear empirical quantification.

**Guidance techniques for condition generations.** Guidance has been shown to play a central role in conditional generation (Dhariwal & Nichol, 2021; Ho & Salimans, 2021), significantly improving alignment between generated samples and conditioning information. More recently, Kynkäänniemi et al. (2024); Karras et al. (2024) proposed techniques to further improve the practical effectiveness of classifier free guidance. Our proposed Noise Awareness Guidance also falls into this category. To the best of our knowledge, it is the first method to explicitly use the noise level itself as a guidance signal, directly enhancing alignment with the intended noise condition.

## 7 CONCLUSION

This paper presents a novel perspective by observing the behavior of the posterior noise level $p_t(t \mid \hat{\mathbf{x}})$, and finds the noise shift issue that the empirically estimated posterior noise level $p_{\phi,t}(t \mid \hat{\mathbf{x}})$ has a tendency toward a larger noise level. We analyze that the noise shift issue is a manifestation caused by a collection of errors from various sources and is widespread in the current denoising sampling process, and performing iterative denoising sampling under noise shifts leads to sub-optimal generations. We further provide a noise awareness guidance approach and its classifier-free variants to directly relieve the noise shift issue and achieve significant improvement by reducing the noise shift gap. We hope that our work will attract researchers to pay attention to the widespread training and inference misalignment in denoising generation and facilitate many possible future research directions, including theoretical or empirical analysis on the noise shift issue, building generative models that are robust to inference shift in sampling stages, exploring the boundary of high-quality generation, or faster sampling.

## REPRODUCIBILITY STATEMENT

To ensure the reproducibility of our results, we provide comprehensive details, including model configurations, key hyperparameters, fine-tuning strategies, and the checkpoints used in Section 5 and Appendix B. We believe that, with these details, the main results can be reproduced with only a few lines of modification to the official DiT and SiT codebases. For the posterior estimator $g_\phi$, which requires additional modifications . Code is publicly available at https://github.com/KlingAIResearch/noise-awareness-guidance.

## ACKNOWLEDGMENTS

This work was supported by Kuaishou Technology, Natural Science Foundation of China (U2342217), Fundamental and Interdisciplinary Disciplines Breakthrough Plan of the Ministry of Education of China (JYB2025XDXM803), Beijing Scholar Program, National Engineering Research Center for Big Data Software.

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

# A  DERIVATION OF STATEMENT 1

We derive the expected noise shift $\delta$ in the presence of additive Gaussian error.

Recall that the forward process is defined for a noise level $t \in [0, T]$ as

$$\mathbf{x}_t = \alpha_t \mathbf{x}_0 + \sigma_t \boldsymbol{\epsilon}, \quad \text{where} \quad \boldsymbol{\epsilon} \sim \mathcal{N}(\mathbf{0}, \mathbf{I}). \tag{12}$$

**Influence of error e.**  Consider an intermediate state perturbed by additive error:

$$\hat{\mathbf{x}}_t = \mathbf{x}_t + \mathbf{e}, \tag{13}$$

where $\mathbf{e} \in \mathbb{R}^D$ is assumed to follow a zero-mean Gaussian distribution with unknown variance, $\mathbf{e} \sim \mathcal{N}(\mathbf{0}, \sigma_e^2 \mathbf{I})$.

The perturbed state can be rewritten as

$$\hat{\mathbf{x}}_t = \alpha_t \mathbf{x}_0 + (\sigma_t \boldsymbol{\epsilon} + \mathbf{e}). \tag{14}$$

Since $\boldsymbol{\epsilon}$ and $\mathbf{e}$ are independent zero-mean Gaussians, their weighted sum is also Gaussian with variance

$$\text{Var}(\sigma_t \boldsymbol{\epsilon} + \mathbf{e}) = \sigma_t^2 \mathbf{I} + \sigma_e^2 \mathbf{I} = (\sigma_t^2 + \sigma_e^2)\mathbf{I}. \tag{15}$$

Thus, the distribution of $\hat{\mathbf{x}}_t$ is

$$\hat{\mathbf{x}}_t \sim \mathcal{N}\big(\alpha_t \mathbf{x}_0, \, (\sigma_t^2 + \sigma_e^2)\mathbf{I}\big). \tag{16}$$

The perturbed state $\hat{\mathbf{x}}_t$ can be expressed in terms of the initial data $\mathbf{x}_0$:

$$\hat{\mathbf{x}}_t = (\alpha_t \mathbf{x}_0 + \sigma_t \boldsymbol{\epsilon}) + \mathbf{e} = \alpha_t \mathbf{x}_0 + (\sigma_t \boldsymbol{\epsilon} + \mathbf{e}). \tag{17}$$

**Definition of noise shift.**  This distribution coincides with that of an intermediate state from the original forward process but evaluated at a shifted noise level $t' = t + \delta$. By definition, $\delta$ satisfies

$$\sigma_{t+\delta}^2 = \sigma_t^2 + \sigma_e^2, \tag{18}$$

and the noise shift is defined as

$$\delta = t' - t. \tag{19}$$

**First-order approximation.**  Assume that $\sigma_t$ is differentiable in $t$ and that the error variance $\sigma_e^2$ is small, so that $\delta$ is also small. A first-order Taylor expansion of $\sigma_{t+\delta}$ around $t$ gives

$$\sigma_{t+\delta} \approx \sigma_t + \dot{\sigma}_t \, \delta, \tag{20}$$

where $\dot{\sigma}_t = \frac{d\sigma_t}{dt}$.

By construction, $\sigma_{t+\delta} = \sqrt{\sigma_t^2 + \sigma_e^2}$. Substituting yields

$$\sigma_t + \dot{\sigma}_t \, \delta \approx \sqrt{\sigma_t^2 + \sigma_e^2}. \tag{21}$$

**Result.**  Solving for $\delta$ gives the following approximation for the noise shift:

$$\delta \approx \frac{\sqrt{\sigma_t^2 + \sigma_e^2} - \sigma_t}{\dot{\sigma}_t}. \tag{22}$$

# B  IMLEMENTATIONS

All experiments are conducted in PyTorch, based on the official DiT (Peebles & Xie, 2023) and SiT (Ma et al., 2024) codebases.

## B.1  IMPLEMENTATION TO MAIN RESULTS

**Architecture configurations.**  We follow the transformer architectures defined in DiT, using four different configurations for various model sizes: Small (S), Base (B), Large (L), and XLarge (XL). All models employ a patch size of 2, and latent states are obtained using the pre-trained Stable Diffusion tokenizer (Rombach et al., 2022). The detailed model architectures are provided in Table 3.

Table 3: **Configurations on DiTs and SiTs.**

| configs | S/2 | B/2 | L/2 | XL/2 |
|---|---|---|---|---|
| params (M) | 33 | 130 | 458 | 676 |
| FLOPs (G) | 6.0 | 23.0 | 80.7 | 118.6 |
| depth | 12 | 12 | 24 | 28 |
| hidden dim | 384 | 768 | 1024 | 1152 |
| heads | 6 | 12 | 16 | 16 |
| patch size | $2\times2$ | $2\times2$ | $2 \times 2$ | $2 \times 2$ |
| latent encoder | SD-VAE(Rombach et al., 2022) | | | |

**Sampler.** For DiT, we directly adopt the DDPM sampler from the official implementation[4]. For SiT, we use the Euler–Maruyama sampler from its official implementation[5], with the default setting $w_t = \sigma_t$ in Equation 5, and the final step size set to 0.04.

**Guidance weights.** For all baselines with CFG, we keep the setting consistent with the original results, using $w_{cfg} = 1.5$. For all results of NAG without CFG, we use $w_{nag} = 3.0$ by default. For NAG combined with CFG, we set $w_{cfg} = 1.2$ and $w_{nag} = 2.0$ by default.

**Training configurations.** We retain most training configurations from DiT and SiT (Peebles & Xie, 2023; Ma et al., 2024), without modifying decay schedules, warmup schedules, AdamW hyperparameters, or applying additional data augmentation or gradient clipping. All results are reported using an exponential moving average (EMA) of model weights with a decay of 0.9999. Our training setup includes two scenarios on ImageNet: (1) training from random initialization (Figure 3); and (2) fine-tuning off-the-shelf pre-trained models (1400 epochs) with an unconditional noise branch (Table 1). Detailed configurations are summarized in Table 4.

Table 4: **Training Configurations on ImageNet**

| configs | from scratch (Figure 3) | fine-tuning (Table 1) |
|---|---|---|
| training iterations | 400K | 50K |
| batch size | 256 | 256 |
| optimizer | AdamW | AdamW |
| $((\beta_1, \beta_2)$ | (0.9,0.999) | (0.9,0.999) |
| noise dropout | 10% | 20% |
| learning rate | $1 \times 10^{-4}$ | $1 \times 10^{-5}$ |

**Fine-tuning with noise condition dropout on ImageNet.** Compared to training from scratch, fine-tuning requires more careful handling to avoid catastrophic forgetting of learned generative capability. Following the strategy for class-unconditional inputs, we introduce a pseudo noise level (i.e., 1001 for DiT, 1.001 for SiT) that remains consistent across inputs, rather than discarding noise embeddings directly. In addition, we reduce the learning rate to one tenth of the original value ($1 \times 10^{-5}$ instead of $1 \times 10^{-4}$) and double the noise dropout ratio to 20%. When training from scratch, the choice of unconditional implementation has only a minor effect on training dynamics.

**Fine-tuning on new datasets.** We strictly follow the setup in Domain Guidance (Zhong et al., 2025), using a constant learning rate of $1 \times 10^{-4}$ and a batch size of 32 with the AdamW optimizer for 24,000 iterations across all datasets. For NAG, we apply 10% noise dropout.

**FID calculation.** For fair comparison across benchmarks, we strictly follow the FID calculation protocol used in the original implementation of each task. For ImageNet generation, we compute

---

[4]https://github.com/facebookresearch/DiT

[5]https://github.com/willisma/SiT

FID scores between generated images (10K or 50K) and all available real images in the ImageNet training set, using ADM's TensorFlow evaluation suite[6] (Dhariwal & Nichol, 2021). For fine-tuning experiments on downstream datasets, we observe small performance variations between different FID implementations. To ensure consistency with results reported in (Zhong et al., 2025), we compute FID scores using a PyTorch implementation[7], comparing 10K generated images against all available images in the test set for each downstream task.

### B.2 IMPLEMENTATION OF EMPIRICAL POSTERIOR ESTIMATOR $g_\phi$.

To empirically identify the noise shift issue, we rely on an external posterior estimator $g_\phi$. Here we describe the construction of the estimator $g_\phi$ used in Section 3 and Section 5.3. Code is publicly available at https://github.com/KlingAIResearch/noise-awareness-guidance.

To reduce computational costs, we fine-tune the existing SiT-XL/2 checkpoint (the same model used for ImageNet generation) by replacing its final layer with a noise level regressor. The regressor is implemented as a two-layer MLP applied to the globally averaged token: the first layer projects the hidden state from 1152 to 576 dimensions with SiLU activation (Elfwing et al., 2018), and the second layer outputs the predicted noise level.

We inherit the training pipeline and hyperparameters from the noise-condition fine-tuning setup on ImageNet described in Section B.1, including a learning rate of $1 \times 10^{-5}$, the same batch size, AdamW optimizer settings, and identical data preprocessing. The key difference is that the noise level is used as the prediction target rather than as an input condition. The model parameters $\phi$ are optimized by minimizing the $L_2$ loss between the predicted and true noise levels, with the noise condition input masked by a pseudo condition (set to 1.001 in practice).

The posterior model operates in the latent space obtained from the SD-VAE (Rombach et al., 2022), avoiding the need to transform noisy latent states back to image space. We train $g_\phi$ on ImageNet $256 \times 256$ for 40 epochs (approximately 200K iterations), reaching a training loss of 0.0002. No EMA is applied to $g_\phi$.

All probability density functions in this paper are plotted using kernel density estimation (KDE) with 5,000 samples.

The samples are constructed in two steps. First, we randomly sample 5,000 images from ImageNet and generate 5,000 noise samples. We then linearly interpolate the images and noise following the linear schedule, producing 5,000 forward trajectories in which intermediate states share the same clean data point $\mathbf{x}_0$ and noise $\epsilon$. Second, we generate 5,000 reverse trajectories using the Euler–Maruyama SDE solver with 20 steps, incorporating the same class information, and save all intermediate states. In both cases, intermediate states within the same trajectory are tied to the same clean data point and noise. Finally, we compute the densities via KDE for samples associated with the same prior $t$ and the same generation process.

## C COMPARISON WITH REPRESENTATIVE BASELINES ON EXPOSURE BIAS

In this section, we compare NAG with several prior works that also aim to address training–inference misalignment. The baselines include two training-free approaches, Epsilon Scaling (ES) (Ning et al., 2023a) and Time Shift (TS) (Li et al., 2023), as well as one training-intensive approach, Input Perturbation (IP) (Ning et al., 2023b). The results are provided in Table 5. NAG significantly outperforms all prior works, especially under the w/o CFG setup.

All these prior works rely on the assumption that model predictions during sampling tend to exhibit larger variance compared to those during training, and they apply manually designed matching strategies to compensate for this. ES scales the model predictions by a fixed hyperparameter, which may not be effective across all sampled states. TS searches for nearby time steps that better align with the inherent variance of the intermediate states, but this approach is sensitive to the number of

---

[6]https://github.com/openai/guided-diffusion/tree/main/evaluations
[7]https://github.com/mseitzer/pytorch-fid

Table 5: **Full comparison on ImageNet 256 × 256 with DiT-XL/2 and SiT-XL/2.** We fine-tune off-the-shelf DiT-XL/2 and SiT-XL/2 checkpoints for 10 additional epochs to support NAG sampling, with and without classifier-free guidance (CFG), following the original setups (Peebles & Xie, 2023; Ma et al., 2024). All metrics are reported on 50k generated images.

| Model | Training Epochs | Generation w/o CFG | | | Generation w/ CFG | | |
|---|---|---|---|---|---|---|---|
| | | FID | Prec. | Rec. | FID | Prec. | Rec. |
| DiT-XL/2 (Peebles & Xie, 2023) | 1400 | 9.62 | 0.67 | 0.67 | 2.27 | **0.83** | 0.57 |
| + ES (Ning et al., 2023a) | 1400 | 12.25 | 0.63 | **0.69** | 2.20 | 0.79 | 0.60 |
| + TS (Li et al., 2023) | 1400 | 13.14 | 0.64 | 0.62 | 3.70 | 0.75 | 0.60 |
| + IP (Ning et al., 2023b) | 10+(1400*) | 10.20 | 0.63 | **0.69** | 2.19 | 0.79 | **0.61** |
| **+ NAG (ours)** | 10+(1400*) | **2.55** | **0.79** | 0.60 | **2.14** | 0.80 | **0.61** |
| SiT-XL/2 (Ma et al., 2024) | 1400 | 8.61 | 0.68 | 0.67 | 2.06 | **0.82** | 0.59 |
| + ES (Ning et al., 2023a) | 1400 | 8.70 | 0.67 | **0.68** | 1.96 | 0.81 | 0.61 |
| + TS (Li et al., 2023) | 1400 | 8.65 | 0.67 | **0.68** | 1.94 | 0.81 | 0.61 |
| + IP (Ning et al., 2023b) | 10+(1400*) | 8.06 | 0.68 | 0.67 | 1.95 | 0.81 | 0.59 |
| **+ NAG (ours)** | 10+(1400*) | **2.26** | **0.75** | 0.66 | **1.72** | 0.77 | **0.66** |

sampling steps and the search window. IP introduces Lipschitz regularization by injecting additional noise into training samples, aiming to reduce the generalization error when $x_t$ becomes misaligned with the conditioning input $t$. However, accumulated errors arise from multiple factors, not solely from model prediction errors. We believe that NAG benefits from avoiding explicit assumptions about the error structure and from being naturally adaptive throughout the sampling process.

The implementations for DiT-XL/2 of these prior works are based on their official repositories. Since these methods are not directly available for flow models, the results on SiT-XL/2 are obtained by reproducing their basic discrete implementations and extending them to continuous setups in our own codebase.

## D    MORE VISUALIZATION RESULTS WITH KERNEL DENSITY ESTIMATION

In this section, we provide the full probability density results of the estimated posterior $t$, as an extension of Figure 1 and Figure 5.

## E    THE USE OF LARGE LANGUAGE MODELS (LLMS)

We use large language models (LLMs) only as grammar checkers during paper writing.

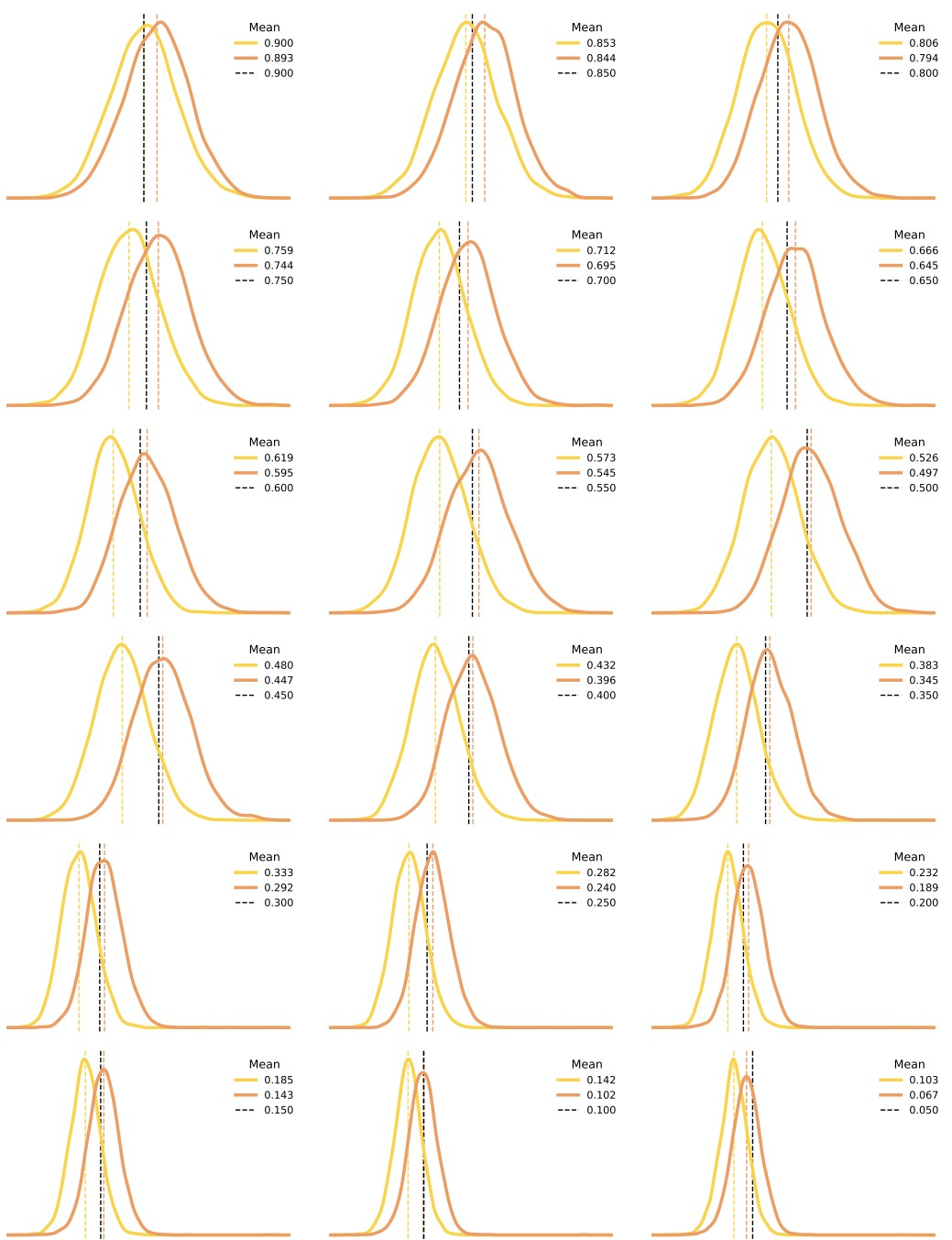

Figure 7: **More visualization of noise shift.** The yellow curves indicate the estimated probability density of the posterior $p_{\phi,t}(t \mid \hat{\mathbf{x}})$ for sampled intermediate states $\hat{\mathbf{x}}$, while the orange curves indicate the posterior $p_{\phi,t}(t \mid \mathbf{x})$ for intermediate states $\mathbf{x}$ stochastically interpolated from training data $\mathbf{x}_0 \sim p_{\text{data}}(\mathbf{x}_0)$ on ImageNet. The black indicator is the pre-defined $t$.

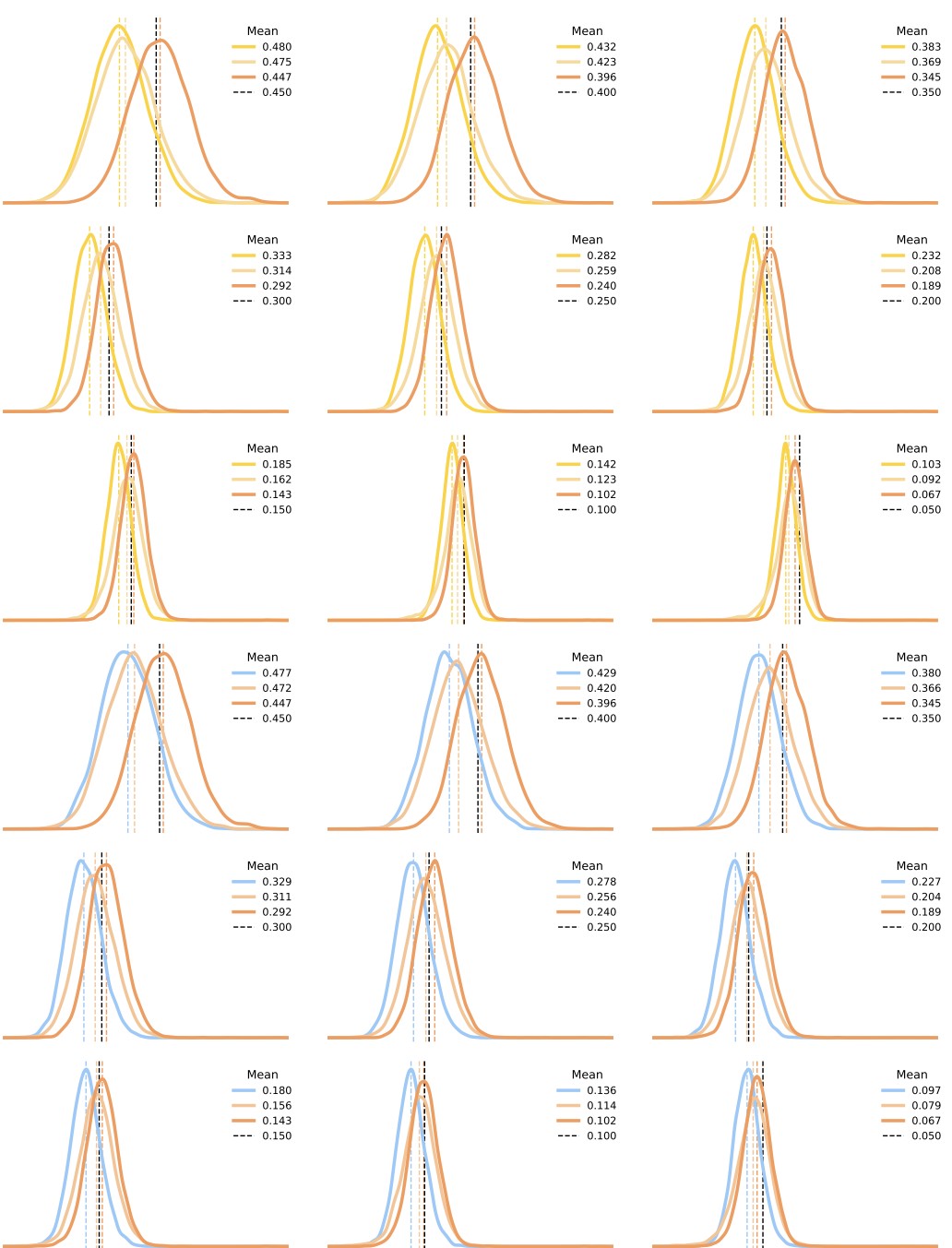

Figure 8: **Additional visualization of how NAG mitigates noise shift.** The yellow curves represent the estimated probability density of the posterior $p_{\phi,t}(t \mid \hat{\mathbf{x}})$ for sampled intermediate states $\hat{\mathbf{x}}$. The blue curve shows the density influenced by CFG, while the pale gold curve highlights the mitigating effect of NAG. The orange curves correspond to the posterior $p_{\phi,t}(t \mid \mathbf{x})$ for intermediate states $\mathbf{x}$ stochastically interpolated from training data $\mathbf{x}_0 \sim p_{\text{data}}(\mathbf{x}_0)$ on ImageNet. The black indicator denotes the pre-defined $t$.

