# OpenReview forum: "Mitigating Noise Shift in Denoising Generative Models with Noise Awareness Guidance"
_ICLR.cc/2026/Conference — ICLR 2026 Poster_

### Official Review · Reviewer_Uw2P · 2025-10-18

**Soundness:** 3
**Presentation:** 2
**Contribution:** 2
**Rating:** 6
**Confidence:** 4

**Summary:**

This paper addresses the issue of  noise shift, a discrepancy in denoising generative models where the actual noise level encoded in an intermediate state deviates systematically from the pre-defined noise schedule.  The authors provide empirical evidence for this shift using an external noise estimator and argue that it leads to sub-optimal generation. To mitigate this, they propose Noise Awareness Guidance (NAG), a training-based correction method that directs the sampling trajectory to remain consistent with the intended noise level.   The method is evaluated on DiT and SiT models for the ImageNet-256 dataset and various supervised fine-tuning tasks, demonstrating competitive improvements in FID.

**Strengths:**

1. The paper discusses and provides some empirical evidence for the Noise Shift.

2.  The approach is based on using the noise level $t$ as a conditioning signal for guidance, drawing an analogy to classifier guidance.

3.  Empirical results indicate that NAG can complement existing guidance (e.g., CFG) and shows potential for improving FID scores on DiT/SiT models.

**Weaknesses:**

1. The paper lacks a thorough comparison with recent methods (e.g., [1]).  The authors incurred the cost of training a specialized component (NAG) and only validated it using basic samplers (DDPM and Euler-Maruyama), but failed to validate its benefit on advanced solvers.

2. Experimental results suggest that the classifier-free NAG variant improves noise estimator accuracy, but it doubles the computational cost by requiring two forward passes per sampling step. Additionally, while the manuscript briefly covers the noise estimator’s training, it lacks a thorough analysis of its robustness, especially for out-of-distribution samples.

3.  *Writing Quality*:  The manuscript requires significant proofreading. For instance,  in the main text, numerous typos include:
     -  "*pespetive*" (Line 475) should be "*perspective*",

     -  "*apporach*" (Line 480) should be "*approach*",

     -   "*varients*"  (Line 480)  should be "*variants*",

     -  "*posible*" (Line 483) should be "*possible*",

     -  "*nosie*"  (Line 476) should be "*noise*",

     -  "*We anaylsis*"  should be "*We analyze*", ....


[1]. Abuduweili, A., et al., Enhancing Sample Generation of Diffusion Models using Noise Level Correction,  TMLR 2025.

[2]. Ning, M., et al.,  Elucidating the Exposure Bias in Diffusion Models, ICLR 2024.

[3]. Lin, S., et al.,  Common diffusion noise schedules and sample steps are flawed, WACV 2024 .

[4]. Tang, Z.,  et al., Inference-Time Alignment of Diffusion Models with Direct Noise Optimization, ICML 2025.

**Questions:**

1. When the prior noise schedule changes (e.g., from uniform to logSNR), does the model need to be retrained to adapt to the new sampling conditions?

2. Is addressing noise shift alone sufficient? How does the method distinguish between noise shift caused by the inherent properties of the sampling process and noise shift arising from external factors, such as approximations, numerical discretization, or model-specific biases?

3. I'm curious whether NAG could help with the numerical instability issue that arises from the schedules in diffusion models, especially during the sampling process as $\sigma_t$ approaches 0?

---

> ### Author Response · Authors · 2025-11-23
> **Response to Uw2P [1/2]**
>
> Thank you for your effort in reviewing our paper. Our responses to your concerns are summarized below.
>
> > W1: The paper lacks a thorough comparison with recent methods
>
> Thank you for pointing out this concern. We fully agree that incorporating additional related work helps clarify our contributions. In the revised version, we have expanded the **Related Work** section by adding a new paragraph titled **Training and inference discrepancy in generative models**, where we include and discuss the relevant studies suggested by the reviewer.
>
> Furthermore, we added a direct comparison with Epsilon Scaling [1].
> We include the additional FID-50k results below (a meaningful complement to Table 1):
> | Model         | FID 50k (w/o CFG) |
> |---------------|------------------ |
> | DIT-XL/2      |  9.26             |
> | DiT-XL/2-ES(b=1.004)  |  12.25    |
> | DiT-XL-NAG    | **2.59**          |
>
> The results show that NAG provides a much more significant improvement in generation quality. We believe this is due to the powerful, flexible, and input-adaptive nature of NAG. In contrast, ES relies on pre-estimated hyperparameters that are applied throughout sampling, which lacks flexibility and does not consider the actual intermediate states.
>
> Notably, ES is highly tied to the specific forward and reverse processes, while our observation and NAG are agnostic to the diffusion formulation and can be easily plugged into both diffusion and flow models, just as CFG does. We sincerely appreciate your suggestions to enhance our paper.
>
> > W1-2 The authors incurred the cost of training a specialized component (NAG) and only validated it using basic samplers.
>
> We respectfully disagree that NAG incurs the cost of training a specialized component. We believe the actual training cost of NAG has been overestimated. Since classifier-based guidance techniques suffer from drawbacks such as high computational cost for training an external posterior estimator, increased pipeline complexity, and the risk of adversarial-like behavior in explicit classifiers, we do not recommend applying classifier-based NAG with a noise-level estimator. In this paper, we only use the noise-level estimator as an empirical noise shift detector to provide intuitive explanation and analysis of NAG.
>
> For classifier-free NAG, we directly embed the dropout operation into the CFG training process without extra cost. We provide fine-tuning scenarios in Tables 1 and 2, and training-from-scratch scenarios in Figure 3. These empirical evaluations cover the two main practical scenarios:
> 1) If we want to keep the model, we can fine-tune the noise-unconditional branch for only an extra 0.7\% training cost.
> As for classifier-free NAG, we can directly embed the dropout operation in CFG training process without extra cost. We provide fine-tuning senarios in Table 1,2 and training from scratch senarios in Figure 3. These empirical evaluation covers the mainly two practice senarios:
> 1) If we want to keep the model, we can fine-tuning the noise unconditional part for extra 0.7\% training cost.
> 2) If we want to continue training or train from scratch, NAG introduces no extra cost beyond dropping some ratio of noise conditions in the inputs.
>
> We appreciate the suggestion to evaluate NAG on advanced samplers. Considering that some advanced solvers such as DPM-Solver may struggle with guidance-based techniques, we believe this is a valuable extension for future investigations. In this paper, we focus on the noise shift issue and the proposed NAG.
>
> > W2-1:Experimental results suggest that the classifier-free NAG variant improves noise estimator accuracy
>
> We cannot say that classifier-free NAG improves noise estimator accuracy, as the noise estimator is only an objective observer used to analyze how NAG influences noise shift.
>
> > W2-2:  but it doubles the computational cost by requiring two forward passes per sampling step
>
> A double forward pass is a common limitation shared by CFG-like approaches. In the default settings of our paper, we do not introduce any newly proposed tricks for CFG to keep the paper clean. These improvements can be promising future work for NAG (such as VGT [2], which increases training budget to reduce guidance sampling cost to one forward pass).

---

> ### Author Response · Authors · 2025-11-23
> **Response to Uw2P  [2/2]**
>
> > W2-3: Additionally, while the manuscript briefly covers the noise estimator’s training, it lacks a thorough analysis of its robustness, especially for out-of-distribution samples.
>
> Since the noise estimator only serves as an objective observer of noise shift and does not directly affect NAG, we did not provide further robustness analysis. In practice, for empirical analysis, we do not obtain the optimal noise estimator (we use the one with a training loss of 0.0002 due to limited resources). Its robustness can be observed from the $p_\text{data}$ curve in Figure 4(a). Moreover, our empirical analysis already accounts for the estimator’s fitness error, as shown by the difference between $p_\phi(t \mid \hat{\mathbf{x}})$ and $p_\phi(t \mid \mathbf{x})$ in Figure 4(b).
>
> > W3: Writing Quality
>
> Thank you for your effort in reviewing again; we have fixed the typos accordingly.
>
> > Q1: When the prior noise schedule changes (e.g., from uniform to logSNR), does the model need to be retrained to adapt to the new sampling conditions?
>
> Since classifier-free NAG only relies on the unconditional noise predictor and shares weights with the main model, whether retraining is required does not depend on NAG, but on the base model.
> Generally:
> - If 'prior noise schedule changes' refers to changing the forward process, then yes, the entire diffusion model must be retrained.
> - If it refers to changing only the discrete sampling schedule, then retraining is not needed.
>
> > Q2:  Is addressing noise shift alone sufficient? How does the method distinguish between noise shift caused by the
> inherent properties of the sampling process and noise shift arising from external factors, such as approximations, numerical discretization, or model-specific biases?
>
> As clarified in Section 5.3, eliminating the estimated noise shift is not a sufficient condition for achieving optimal generation. Since noise shift is a basket of errors, we cannot distinguish the specific error sources. Nevertheless, the presence of a distinguishable noise shift during sampling is a sufficient condition for sub-optimal generation, which motivates addressing the issue directly.
>
>
> > Q3:I'm curious whether NAG could help with the numerical instability issue that arises from the schedules in
> diffusion models, especially during the sampling process as $\sigma_t$ approaches 0?
>
> NAG is only designed to increase noise-level consistency and does not explicitly address numerical instability. As $\sigma_t \to 0$, our empirical analysis also shows that the noise estimator cannot produce reliable results (Figure 4(a)). This is also the region where NAG is less effective.
>
>
> ---
>
> Thanks again for your great effort in reviewing our submission. If our response addresses your concerns, could you please consider updating your assessment and raising the score? If there are still concerns we may have missed, please let us know.

---

### Official Review · Reviewer_zvzx · 2025-10-30

**Soundness:** 3
**Presentation:** 3
**Contribution:** 3
**Rating:** 6
**Confidence:** 3

**Summary:**

This paper identifies a misalignment between the noise levels of the intermediate sampling states and those of the ground-truth data used during training. The authors demonstrate that this issue is prevalent in off-the-shelf diffusion models, highlighting a new direction for improving sampling performance. To address this misalignment, they propose NAG, a guidance technique analogous to classifier-free guidance, designed to correct the sampling trajectory. Empirical results validate the effectiveness of the proposed method.

**Strengths:**

1. The paper is generally well-written and easy to follow.
2. Improving the sampling quality by mitigating the misalignment of the noise level between training and sampling appears new to me.
3. The proposed method is simple to implement and can be easily applied to an off-the-shelf pre-trained model.
4. According to the provided empirical results, the proposed method is very effective.

**Weaknesses:**

1. The paper lacks an ablation study on the choice of $w_\text{nag}$, which makes it difficult for readers to gain a comprehensive understanding of the proposed method.
2. It is known that classifier-free guidance (CFG) trades sample diversity for sample quality. Since the paper claims that the proposed method mitigates noise-level misalignment and provides an effect "orthogonal" to CFG, the results would be more convincing if an ablation study were included to analyze how $w_\text{nag}$ influences sample diversity.
3. (Minor) The work is mainly empirical and lacks theoretical support.

**Questions:**

Please refer to the weaknesses box.

---

> ### Author Response · Authors · 2025-11-23
>
> Thank you for your recognition and valuable suggestions on our work. Our responses are summarized as follows.
>
> > W1&W2: ablation study on the choice of $w_\text{nag}$
>
> Thank you for your valuable suggestions. We have added this ablation study in the newly added Section 5.4. The results can be found in Figure 6(a–b) **in the revision**.
>
> Figure 6(a) shows that, as a guidance-based technique, NAG exhibits a similar effect to CFG. Specifically, we can choose an optimal guidance weight to achieve better generation quality. As $w_\text{nag}$ increases, diversity is sacrificed, which can be viewed as a form of temperature control in denoising generation, even though NAG does not incorporate any class information. This interesting tendency can be regarded as isolating the quality-improvement effect from the class-discrimination effect seen in CFG, and it sheds light on future investigation into how guidance mechanisms improve generation quality.
>
>
> > W3: lacks theoretical support.
>
> Since noise shift is a collection of errors from multiple sources, it is very challenging to conduct deeper theoretical analysis. For NAG, following the derivation of CFG, we intuitively designed the classifier-free version. Nevertheless, our paper provides sufficient empirical evidence, and the NAG approach is simple and practical to use.
>
> ---
>
> Thanks again for your great effort in reviewing our submission. If our response addresses your concerns, could you please consider updating your assessment and raising the score? If there are still concerns we may have missed, please let us know.

---

### Official Review · Reviewer_wzAN · 2025-10-31

**Soundness:** 3
**Presentation:** 3
**Contribution:** 2
**Rating:** 6
**Confidence:** 2

**Summary:**

This paper addresses the noise shift problem in diffusion models: sampling intermediate states has a mismatch between the pre-defined noise schedule and the actual noise levels. The authors propose Noise Awareness Guidance (NAG) to guide back to correct noise. Through empirical results on ImageNet generation and supervised fine-tuning tasks, the paper demonstrates that NAG mitigates noise shift and improves generative sample quality across mainstream diffusion model architectures.

**Strengths:**

- This paper identifies the noise shift with strong empirical evidence (fig 1) , indicating that the actual posterior noise levels during sampling are systematically biased relative to the intended schedule.
- The proposed NAG is simple and effective for diffusion and flow-matching models. And the compatibility of NAG with other guidance mechanisms (e.g., CFG and DoG) is empirically supported and presented in Table 2 and Figs. 4 and 5.
- Clear writing, good ablations, and reproducible (with settings in Appendices)

**Weaknesses:**

The paper omits discussion against several closely related studies addressing noise misalignment and noise correction in diffusion models, for instance, "Enhancing Sample Generation of Diffusion Models using Noise Level Correction", https://arxiv.org/abs/2412.05488

**Questions:**

What is the computational overhead (parameters, runtime, memory) of adopting NAG in practical sampling/fine-tuning settings? How does this compare to other recalibration techniques?

---

> ### Author Response · Authors · 2025-11-23
>
> Thank you for your valuable suggestions. Our responses to the reviewer’s comments are summarized below.
>
> > W1: The paper omits discussion against several closely related studies.
>
> Thank you for pointing out this concern. We fully agree that incorporating additional related work helps clarify our contributions. In the revised version, we have expanded the **Related Work** section by adding a new paragraph titled **Training and inference discrepancy in generative models**, where we include and discuss the relevant studies suggested by the reviewer.
>
> Furthermore, we added a direct comparison with Epsilon Scaling [1].
> We include the additional FID-50k results below (a meaningful complement to Table 1), showing the significance of our approach.
> | Model         | FID 50k (w/o CFG) |
> |---------------|------------------ |
> | DIT-XL/2      |  9.26             |
> | DiT-XL/2-ES(b=1.004)  |  12.25    |
> | DiT-XL-NAG    | **2.59**          |
>
> As for the mentioned paper [2], which was published in TMLR in June 2025, we discuss this work in our newly added paragraph in the related works. As the closest study, [2] also provides a noise-level perspective of the training–inference misalignment; however, their analysis remains primarily theoretical, lacks clear empirical quantification, and is conducted on relatively small-scale tasks. Unlike our empirical conclusion that the noise shift is systematically larger, [2] does not provide such observations. Moreover, [2] relies on training an external model to correct the noise level, which limits its practicality.
>
> > Q1: What is the computational overhead?
>
> NAG completely shares the same framework as classifier-free guidance (it simply performs conditional guidance on the noise level). Therefore, the computational overhead is the same as in the original model.
>
> In fine-tuning/training-from-scratch scenarios, NAG introduces no extra parameters and no extra computational operations (as we only modify the input to the network by performing dropout on the $t$ condition). Thus, it remains completely identical to the backbone model.
>
> In sampling scenarios, NAG performs the same operations as CFG: double forward passes with and without the noise condition. In the default settings of our paper, we do not introduce any newly proposed tricks in CFG in order to keep the method clean. These improvements can be promising future work for NAG (such as VGT [3], which increases the training budget and reduces guidance-time cost to a single forward pass).
>
> ---
>
> Thanks again for your great effort in reviewing our submission. If our response addresses your concerns, could you please consider updating your assessment and raising the score? If there are still concerns we may have missed, please let us know.
>
> [1] Elucidating the Exposure Bias in Diffusion Models, ICLR 2024
>
> [2] Enhancing Sample Generation of Diffusion Models using Noise Level Correction, TMLR 2025.6
>
> [3] Visual Generation Without Guidance, ICML 2025

---

### Official Review · Reviewer_JnRR · 2025-10-31

**Soundness:** 3
**Presentation:** 3
**Contribution:** 2
**Rating:** 4
**Confidence:** 3

**Summary:**

This paper aims to solve a train-inference mismatch in diffusion models. Specifically, during the sampling procedure, the actual noise encoded in intermediate states drifts to larger noise levels than the intended schedule, which harms generation quality. To address this, the authors propose Noise Aware Guidance (NAG), which adds a guidance term based on the gradient of the posterior noise level to pull trajectories back toward the correct noise manifold; they provide both a classifier-based version using an external noise-level estimator and a classifier-free variant implemented by mixing conditional and unconditional scores via noise-condition dropout.

**Strengths:**

- **Well-written and easy to follow.** The paper clearly motivates the noise-shift train–inference mismatch and steadily builds evidence for it.
- **Principled method.**: NAG is logically derived (using ($\nabla_x \log p(t\mid x)$) to pull states back to the intended noise level), simple to plug in, and compatible with diffusion/flow samplers and with CFG/DoG.
- **Consistent gains.** Across DiT/SiT backbones and multiple datasets, NAG improves FID and reduces the measured noise shift, with ablations that isolate its contribution and clarify when it helps (notably at higher SNR).

**Weaknesses:**

- My major concern is that there are some papers targeting the distribution mismatch between training and inference. In general, it is called exposure bias, and this is well explored in [1]. To summarize that paper, they derive how the variance of the backward diffusion model's process could deviate from the forward diffusion process, and give a solution scaling the estimated score to match the variance. Comparing this submission to [1], the flow is extremely similar: 1) finding the distribution mismatch between training and inference, and 2) solving this mismatch with NAG rather than using the score scaling method in [1]. Considering this similarity, the authors' contribution is that finding and addressing the distribution mismatch between training and inference no longer holds. Furthermore, completely ignoring these works makes me suspicious that the authors investigated related literature well. I recommend comparing [1] and the following papers about exposure bias, and adding these works to the related work section.

- The method should be evaluated across various model sizes. And, I am also curious whether the distribution mismatch trend could be changed according to model size.


Reference

[1] ELUCIDATING THE EXPOSURE BIAS IN DIFFUSION MODELS, ICLR 2024.

**Questions:**

N/A

---

> ### Author Response · Authors · 2025-11-23
>
> We sincerely thank the reviewer for the careful reading of our paper and the insightful suggestions. Our responses to each concern are provided below.
>
> > W1-1: major concern on missing related works on training and inference mismatching
>
> Thank you for pointing out this line of related work. We agree that incorporating these papers into our **Related Work** section helps clarify our contributions more thoroughly. We have added a new paragraph **Training and inference discrepency in generative models** and included the relevant works accordingly in our revision.
>
> > W1-2: the contributions of this work.
>
> We respectfully disagree with your assessment of our contributions based on the paper flow.
> First, the existence of training–inference discrepancy in generative modeling is well known and extensively studied in both autoregressive models and diffusion-based models. We never claim that discovering this mismatch is our contribution.
> Instead, our distinct contributions are:
> 1) **Quantifying the mismatch via an external observer**. We introduce an empirical and objective measure of mismatch via an external posterior estimator $p_\phi(t\mid \mathbf{x})$, which depends solely on the noise ratio in the input and is agnostic to the specific diffusion/flow noise schedule. This is fundamentally different from [1], where exposure bias is analytically defined and tightly coupled to the specific formulation of the model.
> 2) **Reformulating the mismatch as a condition-consistency problem, which directly motivates NAG.**
> Based on the proposed 'noise shift' perspective, we reinterpret the mismatch as a noise-level conditioning inconsistency. This leads naturally to the design of NAG, which directly addresses this conditioning misalignment.
>
> Both components (the empirical characterization + the NAG method) are novel contributions brought by this work.
>
> Regarding the paper structure, we believe the flow is natural: first identify the mismatch, then propose a method that directly targets it. This is a common and acceptable structure. Importantly, our NAG method is specifically motivated by the noise-level conditioning viewpoint, and presenting this first helps readers understand the motivation behind NAG clearly. We do not think that similarity in overall structure diminishes the originality of our contributions.
>
> > W1-3: Compared to the mentioned paper.
>
> To fully address your concern, we add an additional comparison against the method in [1] (Epsilon Scaling, ES) on DiT-XL/2 with their official repository
> https://github.com/forever208/ADM-ES.
>
> We include the additional FID-50k results below (a meaningful complement to Table 1).
> | Model         | FID 50k (w/o CFG) |
> |---------------|------------------ |
> | DIT-XL/2      |  9.26             |
> | DiT-XL/2-ES(b=1.004)  |  12.25            |
> | DiT-XL-NAG    | **2.59**          |
>
> The results show that NAG provides a substantial improvement compared to ES. We believe the reason is that NAG is adaptive, input-dependent, and flexible, while ES relies on pre-estimated static hyperparameters and thus lacks responsiveness to the actual intermediate states encountered during sampling.
>
> Notably, ES is tightly tied to a specific forward/reverse diffusion design. In contrast, our noise-shift observation and NAG formulation are agnostic to the diffusion parameterization and apply to both diffusion and flow models, in the same plug-and-play manner as CFG. We sincerely appreciate your suggestion to include this comparison.
>
> > W2: The method should be evaluated across various model sizes
>
> Thank you for the suggestion. We have included evaluations for four different model sizes in the original Figure 2. Please refer to the results.
>
> We believe your concern may also relate to the question: 'Can better convergence release the noise shift issue?' Unfortuanately, the answer is negative. We provide empirical observation on the effect of the model sizes, and we can see from the results that with the increase of the model size, the observed noise shift remain a relative stable value (Section 5.4 and Figure 6(c) in the revised paper).
>
> Additionally, our main empirical analysis uses SiT-XL/2 trained for 1400 epochs, which is widely believed to be a highly converged ImageNet-256×256 model, yet it still exhibits a significant noise shift.
> This further confirms that noise shift is not mitigated by larger or better-trained models, and supports the necessity of methods such as NAG.
>
> ----
>
> Thank you again for your thoughtful and constructive review.
> If our responses have addressed your concerns, we kindly ask you to reconsider your evaluation and raise the score. Should any concerns remain, we are very happy to provide further clarification.
>
> [1] ELUCIDATING THE EXPOSURE BIAS IN DIFFUSION MODELS, ICLR 2024.

---

> ### Comment · Reviewer_JnRR · 2025-11-25
>
> I sincerely appreciate the authors' response. Thanks to the detailed explanations, some of my concerns have been addressed, while others remain.
>
>
> > W1-1: major concern on missing related works on training and inference mismatching
>
> I appreciate the enriched related work section. In the next revision, I hope the authors can further expand on prior work discussing exposure bias in diffusion/flow models. Additionally, I find that the writing quality issue pointed out by Reviewer Uw2P is also important and should be carefully addressed in a future revision.
>
> > W1-2: the contributions of this work.
>
> 1. **Quantifying the mismatch via an external observer.**: After reading the response, it seems that you agree that the distribution mismatch between training and inference is already widely recognized as an important issue in the community. Therefore, empirically showing this mismatch—without a deeper theoretical or empirical investigation into how it manifests or why it matters—feels somewhat trivial. I feel that a deeper exploration of empirical patterns and mismatches from theories is necessary, which makes an empirical observation meaningful. Without such a deeper analysis, I have difficulty considering this observation a substantial contribution.
>
> 2. **Reformulating the mismatch as a condition-consistency problem, which directly motivates NAG.**: I agree that inference–training mismatch provides motivation for NAG. However, extending the above point, prior work has already observed exposure bias, so reiterating this through relatively simple observations does not seem to add significant novelty.
>
> > W1-3: Compared to the mentioned paper.
>
> I cannot yet conclude that NAG consistently outperforms previous methods for mitigating exposure bias. 1) Please compare against -ES across all setups in Table 1 (including CFG and SiT). 2) Please also include other established approaches for exposure-bias mitigation. Since the authors have already expanded the related work section in the revised draft, I believe conducting these comparisons would be sufficient to evaluate the method’s relative effectiveness. Without these results, I cannot confirm the effectiveness of the proposed NAG over previous approaches.
>
> > W2: The method should be evaluated across various model sizes
>
> I sincerely apologize for the earlier misunderstanding, and thank the authors for the clarification.
>
> ---
>
> If these issues are thoroughly addressed during the rebuttal, I would be willing to consider raising my score.

---

> > ### Author Response · Authors · 2025-11-27
> >
> > We sincerely thank the reviewer for the active discussion. Our additional clarifications to the remaining concerns are as follows:
> >
> > > W1-1: further expand on prior work discussing exposure bias in diffusion/flow models.
> >
> > We further expand the training–inference misalignment paragraph with one newly published paper [2]. We agree that a richer related works section provides a fuller picture of this topic for readers and grounds our paper more solidly.
> > Notably, as far as our investigation shows, prior works using the terminology *exposure bias* only discuss this issue in diffusion models, while in this paper we also incorporate flow-based models.
> > We will continue polishing our writing in future revisions. If you believe some important literature is missing and should be included, please feel free to point it out, we are happy to add it.
> >
> > > W1-2: the contributions of this work.
> >
> > We respectfully disagree with the reviewer's reduction of our contribution.
> >
> > First, observing the training–inference noise-level shift using an external noise estimator is simple and straightforward but still non-trivial, since none of the prior works attempt to reformulate misalignment explicitly as a noise-level shift and quantify it. (For example, the mentioned work [1] (Epsilon Scaling) uses the variance of the model output as an indicator of misalignment.)
> > Second, we believe we have provided the **necessary theoretical derivation** explaining why the sampling noise level $t$ always shifts toward a larger $t'$, causing the observable noise shift $\delta$, **(Section 3 and Appendix A)**. We also provide a discussion on why this shift matters from the perspective of **Lipschitz continuity and shifted SDE computation points (Lines 209–214)**.
> >
> > We believe these contributions should not be subjectively neglected.
> > We also emphasize that the novel methodology NAG, which directly targets mitigating noise-level conditioning misalignment, is another important contribution.
> >
> > > W1-3: Compared to mentioned paper.
> >
> > We appreciate the reviewer's suggestion to compare NAG with more related works, which would indeed further strengthen the paper. Unfortunately, it is not feasible for us to conduct such extensive comparisons within the review cycle. The main obstacles are:
> > 1. The aligned evaluation in Table 1 is quite slow, and these prior works have never been evaluated on convergent models such as the 1400-epoch SiT-XL/2 on ImageNet256.
> > 2. None of the prior works have evaluated their methods on flow-based models (SiT).
> > 3. Constructing these baselines on the (D/S)iT-XL/2 benchmark is very time-consuming and expensive.
> >
> > We would also like to note that **none of the mentioned prior works are required to include comparisons with one another to confirm their effectiveness in their original papers**. As all of these works target the misalignment from distinct perspectives, it is appealing to investigate their collaboration in future work.
> >
> > Nevertheless, in order to fully address your concern, we extend the comparison to ES under all setups in Table 1, following your suggestion. We agree that building this standard benchmark to compare this group of approaches is meaningful for the community. The results are shown below.
> >
> > | Model         | FID 50k (w/o CFG) | FID 50k (with CFG) |
> > |---------------|------------------ | ------------------ |
> > | DiT-XL/2      |  9.26             | 2.27 |
> > | DiT-XL/2-ES(b=1.004)  |  12.25    | 2.19 |
> > | DiT-XL-NAG    | **2.59**          | **2.14** |
> > | SiT-XL/2      |  8.61             | 2.06 |
> > | SiT-XL/2-ES(b=1.004)  |  8.69    | 1.98 |
> > | SiT-XL-NAG    | **2.26**          | **1.72** |
> >
> > It is interesting that ES performs poorly without CFG, while NAG overcomes this setup.
> > We hope this captures the main insight that the reviewer is interested in.
> >
> > Due to the 10-page limitation of the revision, we currently cannot directly merge these results into Table 1. We will reorganize the paper to place these results in the main body in the next revision.
> >
> > [1] Elucidating the Exposure Bias in Diffusion Models, ICLR 2024
> >
> > [2] Anti-Exposure Bias in Diffusion Models, ICLR 2025
> >
> > ----
> > Thank you again for your valuable suggestions. If our additional responses have resolved your concerns, we would be grateful if you could reconsider your evaluation and raise the score accordingly. If any concerns remain, we would be very happy to provide further clarification.

---

> > > ### Comment · Reviewer_JnRR · 2025-11-27
> > >
> > > ### W1-1 & W1-2.
> > >
> > > Thank you very much for the detailed clarifications. However, I would still like to see a more thorough revision that explicitly explains how this work differs from previous studies on exposure bias in diffusion and flow models, and how the authors’ derivation is conceptually and technically connected to those prior works. In the current version, these relationships remain insufficiently clear.
> > >
> > > ---
> > >
> > > ### W1-3.
> > >
> > > Thank you for completing Table 1. I understand that time is limited. However, without concrete experimental comparisons, it is difficult for me to confirm the effectiveness of the proposed NAG over previous approaches for mitigating the distribution mismatch between training and inference.
> > >
> > > As a feasible compromise, I strongly suggest at least conducting comparisons with training-free sampling-based methods. I believe it is reasonable to defer comparisons with training-intensive baselines to a later version. My understanding is also that NAG itself does not require prohibitive additional training cost, so demonstrating its advantage over other training-free methods would already provide meaningful evidence of its practical value.
> > >
> > > ---
> > >
> > > I would like to give it a higher score if the revised version clearly highlights the conceptual differences from prior exposure-bias studies and includes comparisons with a few representative training-free baseline methods. However, in its current form, without these clarifications and comparisons, I regret that I cannot support acceptance.

---

> ### Author Response · Authors · 2025-12-03
> **Additional Response to Revewer JnRR**
>
> Thank you for your effort in actively participating in the discussion phase. After carefully considering your suggestions, we have continued improving our paper accordingly.
>
> The additional improvements are summarized below:
>
> 1. Following your suggestion, we introduce **three** representative baselines in Table 1, including two training-free approaches (Epsilon Scaling (ES) [1] and Time Shift (TS) [2]) and one training-intensive method (Input Perturbation (IP) [3]).
> The full results are included in the Appendix C
>
> | ImageNet  | FID w/o CFG   | prec  w/o CFG | rec  w/o CFG | FID with CFG  | prec with CFG | rec with CFG |
> |-----------|-------|------|------|-------|------|------|
> | DiT-XL/2  | 9.62  | 0.67 | 0.67 | 2.27  | **0.83** | 0.57 |
> | +ES [1]   | 12.25 | 0.63 | **0.69** | 2.20  | 0.79 | 0.60 |
> | +TS [2]   | 13.14 | 0.64 | 0.62 | 3.70  | 0.75 | 0.60 |
> | +IP [3]   | 10.20 | 0.63 | **0.69** | 2.19  | 0.79 | **0.61** |
> | +NAG (ours)| **2.55**  | **0.79** | 0.60 | **2.14**  | 0.80 | **0.61** |
> |-----------|-------|------|------|-------|------|------|
> | SiT-XL/2  | 8.61  | 0.68 | 0.67 | 2.06  | **0.82** | 0.59 |
> | +ES [1]   | 8.70  | 0.67 | **0.68** | 1.96  | 0.81 | 0.61 |
> | +TS [2]   | 8.65  | 0.67 | **0.68** | 1.94  | 0.81 | 0.61 |
> | +IP [3]   | 8.06  | 0.68 | 0.67 | 1.95  | 0.81 | 0.59 |
> | +NAG (ours)| **2.26**  | **0.75** | 0.66 | **1.72**  | 0.77 | **0.66** |
>
> - ES [1] assumes that the variance of model predictions during inference is larger than during training, and scales it back using fixed hyperparameters.
> - TS [2] follows a similar assumption and searches nearby timesteps to better match the variance of intermediate states.
> - IP [3] introduces extra noise in training samples to provide Lipschitz regularization, reducing generalization error when inputs $x_t$ are misaligned with the conditioning input $t$.
>
> Compared to these prior works, NAG does not make explicitly assumption on the format of the misalginemnt, and is adaptive during the sampling process, due to computing the guidance signal by the model on-the-fly. These factor can state why NAG outperforms prior works.
>
> 2. Along with our newly added comparisons, we provide a discussion on how this paper is conceptually and technically connected to these prior works. (Appendix C)
>
> To this end, we have made every effort to resolve your concerns, and we believe the further comprehensive comparisons and discussion indeed make the paper much more solid. We hope these improvements will convince you to assign a higher score.
>
> ------
> [1] Elucidating the Exposure Bias in Diffusion Models, ICLR 2024
>
> [2] Alleviating Exposure Bias in Diffusion Models through Sampling with Shifted Time Steps, ICLR 2024
>
> [3] Input Perturbation Reduces Exposure Bias in Diffusion Models, ICML 2023

---

### Author Response · Authors · 2025-12-03
**Summary of the Discussion Phase**

Dear all,

We sincerely thank the reviewers and the area chair for their hard work in evaluating our submission.

We are encouraged by the recognition of NAG’s novelty (`zvzx`), simplicity (`all reviewers`), and performance (`JnRR`, `wzAN`, `zvzx`). Reviewers also highlighted the paper’s clear writing (`JnRR`, `wzAN`, `zvzx`), good ablations (`JnRR`, `wzAN`), and strong empirical evidence (`JnRR`, `wzAN`, `zvzx`).

The main concerns fall into two categories:

> Concern #1: Limited discussion of related works (`JnRR`, `wzAN`, `Uw2P`)

We expanded the related work section with a new paragraph *Training–inference misalignment*, in which we carefully position the prior studies and clarify their distinctions from our work.

To fully address the concerns raised by `JnRR`, we additionally extended Table 1 to include **three** representative prior works: two training-free approaches and one training-intensive approach. (See Appendix C.)

We clarify that the contributions of this paper include:

1. We provide clear empirical evidence that training–inference misalignment can be directly observed as a systematic drift toward larger posterior noise levels in intermediate states.
2. We propose a conceptually and technically novel approach, *NAG*, which is simple, principled, effective, and easy to implement.

> Concern #2: Need for more ablations (`zvzx`)

We added additional ablation studies in Section 5.4 of the revised manuscript.

We believe all concerns raised by the reviewers have been thoroughly addressed.
We have updated the manuscript accordingly, with revisions highlighted in **blue**.

We are very grateful for the opportunity to refine and strengthen the paper during this process.

---

### Meta-Review · Area_Chair_23pA · 2026-01-07

**Summary:**

Across reviews, the paper is generally seen as well-written, simple to apply, and empirically strong, with consistent gains on ImageNet and across diffusion/flow backbones.
The decision-critical concerns were mostly about positioning/novelty relative to “exposure bias” / training–inference mismatch literature and whether the method is meaningfully different from (and better than) prior correction approaches. Reviewer JnRR in particular pushed hard on this: unclear conceptual connection/differences to prior exposure-bias work and insufficient baseline comparisons (especially training-free ones).
After rebuttal, the authors did the right things: expanded related work and added direct comparisons against representative baselines (ES/TS/IP) showing NAG is substantially better on the reported ImageNet setups. This materially reduces the “maybe it’s just a repackaging” risk.

Recomendation: Accept (poster). The main reason is that the empirical case + added baselines now supports “this works and seems to beat the obvious alternatives,” even if the exposition around conceptual novelty vs exposure-bias literature still isn’t perfectly crisp.

**Reviewer Concerns:**

Addressed in rebuttal:
-	Need for additional ablations (guidance weight / diversity tradeoff): Added ablations and explicitly discussed that increasing the guidance weight sacrifices diversity (CFG-like behavior), which directly answers zvzx’s main request.
-	Related work gaps (noise correction/exposure bias): Reviewers flagged missing closely related studies; authors expanded the related work and discussed the TMLR’25 noise-level correction direction.
-	Baseline comparisons vs exposure-bias mitigation methods: This was the big one. Authors added/extended comparisons against ES and then further added TS and IP as representative baselines, with NAG clearly ahead in the shown results.
-	Writing quality/ typos: Uw2P listed concrete typos; authors claim they fixed them.
-	Compute/overhead question: Authors clarified NAG has CFG-like sampling overhead (two passes) and no extra parameters in training/fine-tuning, which answers wzAN’s question in a concrete way.
Still outstanding/not fully resolved:
-	Conceptual clarity vs prior exposure-bias work (JnRR): Even after discussion additions, JnRR’s underlying point remains: the paper needs a cleaner, more explicit “here is what is the same vs different” story relative to exposure-bias analyses and fixes. JnRR explicitly said the relationships were still insufficiently clear at that stage.
-	Evaluation on advanced samplers/solvers (Uw2P): Authors acknowledge this and essentially defer it, noting that some advanced solvers can be finicky with guidance. This is a reasonable limitation but it remains a gap.
-	Robustness of the external noise estimator: Authors argue it’s only an “observer,” so robustness doesn’t matter much; that’s partly true, but it still underpins several diagnostic plots/claims, so a bit more validation would strengthen confidence.
-	Compute cost in sampling: Even if “same as CFG” is clear, doubling forward passes is still a real practical cost; the rebuttal explains it but does not remove it.

**Reviewer Scores:**

-	Reviewer JnRR (score: 4 marginally below the acceptance threshold). Likely 4 to 6. Originally marginally below threshold, primarily on novelty/positioning and missing comparisons. After the added ES/TS/IP comparisons, the “show me it beats training-free baselines” request is substantially addressed. I’d expect JnRR to move to a reluctant 6 (marginally above the acceptance threshold) assuming the revised manuscript also makes the exposure-bias connection more explicit than the discussion thread suggested. (If not, they might stay at 4, but the new baselines are a big deal.)
-	Reviewer wzAN (score: 6 marginally above the acceptance threshold).  Likely stays 6. Main issues were missing related work and overhead. Both were addressed cleanly, but the reviewer already hedged (“would not mind if rejected”) and had low confidence. So I’d expect stay at 6.
-	Reviewer zvzx (score: 6 marginally above the acceptance threshold).  Likely 6 to 8. Their key asks were the guidance-weight ablation and diversity impact discussion; authors explicitly added this and described the diversity/quality tradeoff.  This is exactly the kind of rebuttal that usually yields a raised score.
-	Reviewer Uw2P (score: 6 marginally above the acceptance threshold). Likely stays 6.  They raised several practical concerns: advanced solvers, cost, estimator robustness, and writing. Writing got fixed, cost was clarified, but advanced-solver validation and robustness weren’t really closed: more “future work” than “done.”  So I’d expect them to hold at 6.

---

### Decision · Program_Chairs · 2026-01-26

Accept (Poster)